



# The effect of 2020 COVID-19 lockdown measures on seismic noise recorded in Romania

Bogdan Grecu[1], Felix Borleanu[1], Alexandru Tiganescu[1,2], Natalia Poiata[1,3], Raluca Dinescu[1], Dragos Tataru[1]

[1]National Institute for Earth Physics, Magurele, 050811, Romania
[2]Technical University of Civil Engineering Bucharest, Bucharest, 020396, Romania
[3]Université de Paris, Institut de Physique du Globe de Paris, CNRS, F-75005 Paris, France

*Correspondence to*: Bogdan Grecu (bgrecu@infp.ro)

**Abstract.** After the World Health Organization declared COVID-19 a pandemic in March 2019, Romania followed the example of many other countries and imposed a series of restrictive measures, including restricting people's mobility and closing social, cultural and industrial activities to prevent the spread of the disease. In this study, we analyze continuous vertical component recordings from the stations of the Romanian Seismic Network - one of the largest networks in Europe containing 148 stations - to explore in detail the seismic noise variation associated with the reduced human mobility and activity in Romania due to COVID-19. We focused our investigation on four frequency bands - 2-8 Hz, 4-14 Hz, 15-25 Hz and 25-40 Hz - and found that the largest reductions in seismic noise associated with the lockdown corresponds to the high frequency range, from 15 to 40 Hz. We found that all the stations with large reductions in seismic noise (> ~40%) are located inside and near schools or in buildings, indicating that at these frequencies the drop is related to the drastic reduction of human activity in these edifices. In the lower frequency range (2-8 Hz and 4-14 Hz) the variability of the noise reduction among the stations is lower than in the high frequency range, and the noise level is reduced by up to 35%. This drop is due to reduced traffic during the lockdown, as most of the stations showing such changes in seismic noise in these bands are located within cities, near main or side streets. In addition to the noise reduction observed at stations located in populated areas, we also found seismic noise lockdown-related changes at several stations located far from urban areas, with movement of people in the vicinity of the station explaining the noise reductions. Apart from the opportunity to investigate in more detail the seismic noise characteristics due to human mobility and activity, we show that noise reduction during the lockdown has also improved the earthquake detection capability of the accelerometers located in noisy urban environments.

## 1 Introduction

Seismic stations record various types of signals, from transients, like earthquakes, to the continuous small amplitude ground vibrations of the Earth, often referred to as seismic noise. The latter has different origins and specific characteristics, depending on the frequency band in which it is recorded. At low frequencies (0.05–0.5 Hz), seismic noise has natural origin being linked to oceanic activities (Longuet-Higgins, 1950; Hasselmann, 1963) and exhibiting strong seasonal variations (McNamara and



Buland, 2004, Marzorati and Bindi, 2006, Diaz et al., 2010, Evangelidis and Melis, 2012, Grecu et al., 2012). Above 1 Hz, the seismic noise is mainly generated by different human and industrial activities presenting a pronounced variability between daytime and nighttime as well as between working-days and weekends. This signature of anthropogenic origin on seismic noise at high frequencies has been investigated and recognized in many studies around the world (McNamara and Buland,

2004, Groos and Ritter, 2009, Diaz et al., 2010, Nakata et al., 2011, Diaz et al., 2017, Grecu et al., 2018).

The year 2020 has witnessed an unprecedented disruption in anthropic activities in many cities around the globe caused by the 2019 coronavirus disease (COVID-19) and having a direct effect on seismic noise recorded by seismic stations. Recent studies have shown a significant reduction of the noise level due to the restrictions imposed by authorities at local, regional and national scales to prevent the spread of COVID-19. The noise reduction was noticed at stations located inside the cities (e.g.,

Tokyo, Barcelona, Milano,Verona, Florence, Catania, Auckland, Querétaro, see Yabe et al., 2020, Diaz et al., 2021, Poli et al., 2020, Cannata et al., 2021, van Wijk et al., 2021, De Plaen et al., 2021), but it has also been reported at stations far from populated areas (e.g., The Black Forest, see Lecocq et al., 2020a). The observed drop in the seismic noise is not uniform, it varies from one place to another depending on the characteristics of the noise sources in the area and the way they were affected by the different societal responses to activity restrictions (Xiao et al., 2020). The most comprehensive overview of the

variations in seismic noise induced by anthropogenic activities during the COVID-19 pandemic is presented by Lecocq et al. (2020a). The study analyzed noise data from more than 300 stations distributed worldwide pointing out that disruptions in human activities such as traffic reduction, school closure or reduction of tourist activities are responsible for the drop in the high-frequency (4–14 Hz) seismic noise levels of up to 50%.

Similar to other countries, Romania has been significantly affected by the COVID-19, the first official case in the country

being reported on February 26, 2020. After the spread of the virus became an undisputed reality, the government has started imposing mobility restrictions to limit the transmission of the virus and the number of infections. These restrictive measures were imposed gradually and the first ones were taken on March 11, 2020 when all schools in Romania were closed. On March 16, 2020 the state of emergency was declared and the next day the first military order was issued. This order led to banning all outdoor activities, the closure of cafes and the restriction of the number of people in outdoor activities to a maximum of 100

persons. On March 21, 2020, the second military order was issued. It led to the closure of all shopping centers, banning of groups of more than 3 people in the streets during daytime and imposed the curfew from 10 p.m. to 6 a.m. The Romanian government instituted the national lockdown on March 24, 2020 when all movement was restricted, except for work purposes, health needs and essential activities. The lockdown ended on May 14, 2020, and starting with May 15, 2020 gradual relaxation measures (opening of some shops, museums, etc.) were resumed. Since then, no other lockdown has been imposed in Romania.

On July 18, 2020 the quarantine law came into force. This law regulated some necessary measures that ceased with the lifting of the state of emergency, but still needed for limiting the spread of the COVID-19.

The emergency measures taken to prevent the spread of the COVID-19 in Romania provided a good opportunity to investigate the changes of seismic noise levels in relation to human activities during the restriction period (March 11 - May 15, 2020). In





this study, we analyse the continuous data recorded at seismic stations of the Romanian Seismic Network (RSN) to reveal
seismic noise variations before, during and after the Romanian lockdown.

## 2 Data and method

The RSN has been permanently enhanced and enlarged in the last two decades, becoming one of the largest real-time seismic
networks in Europe (Marmureanu et al., 2021). RSN is operated by the National Institute for Earth Physics (NIEP) and consists
of 148 stations (as of September 2020) equipped with strong and weak motion instruments. Of these, 43 stations have
accelerometer sensors, 76 stations include both accelerometers and broadband velocity sensors, while 29 stations have
accelerometers collocated with short-period velocity sensors (Figure 1). All stations record continuously the ground motion at
100 Hz sampling rate and the data are transmitted in real-time to the NIEP's Data Center. The RSN stations are deployed all
over Romania in different environments. Of the 148 stations, 32 are installed in remote areas, 31 in sparsely populated places
(population less than 2000 inhabitants), 24 in villages, communes or small cities with a population between 2000 and 10000
inhabitants, while 61 stations are installed within the medium to large urban areas (Figure 1).

To study the seismic noise variations, we analyse the continuous recordings from the vertical component of velocity and
accelerometer sensors of the RSN stations that cover the time period from March 4, 2019 to September 27, 2020. The data
processing was performed following the approach described by Lecocq et al. (2020a) and using the publicly available
SeismoRMS software package (Lecocq et al., 2020b). The probabilistic power spectral density (PPSD) acceleration amplitudes
(McNamara and Buland, 2004) are computed for each day using 1800s time windows with 50% overlap. The PPSDs are then
converted to displacement spectral powers and finally, using Parseval's identity, to the displacement root-mean-square
(DRMS) in the frequency domain of interest.

We investigate the changes in the DRMS in 4 frequency bands: 2–8 Hz, 4–14 Hz, 15–25 Hz and 25-40 Hz (hereinafter referred
to as Band 1, Band 2, Band 3 and Band 4, respectively). We choose the above frequency intervals taking into account different
contributions that the anthropogenic noise sources have in a wide frequency range, starting from 0.02 Hz up to 40 Hz (Sheen
et al., 2009; Boese et al., 2015; Diaz et al., 2017).

We present the results obtained mainly from the analysis of the accelerometer recordings, because many stations in urban areas
have only strong motion instruments and in addition, the DRMS' computed from either accelerometers or velocity sensors
show similar shape for all the investigated frequency bands (see Figure S1 from Supporting information (SI) section).
However, for one particular station (MLR), the noise reduction was investigated on the broadband velocity sensor.

With the crisis caused by the COVID-19, Google and Apple have made available to authorities and general public the mobility
data they have used in the map products to mitigate the effect of the emergency situation. To analyse the potential connections
between seismic noise variation and mobility data across Romania, we used both data sets (Google and Apple mobility data)
for two-time intervals, February 15 - September 27, 2020 and January 13, 2020-September 27, 2020, respectively.



## 3 Results

### 3.1 General overview

Given the significant differences in terms of the locations of the seismic stations, we first quantitatively assessed the level of noise reduction across all the RSN stations during the national lockdown. For each station and defined frequency band, we computed the median of the noise DRMS for two intervals of 30 days, one ending just before the school closure in Romania (from February 10 to March 10, 2020) and one during the lockdown (from March 25 to April 23, 2020). We compute the percentage of noise reduction between the two intervals and display it on the map, in Figure 2. For each site, we represented a circle colored according to the maximum percentage of the noise reduction in each band and sized as a function of the number of inhabitants in the area (Figure 2).

We observe noise reduction for all frequency bands. Band 1 shows the most homogeneous distribution of the noise reductions and the smallest variability of their values. A large number of stations (46) show noise reduction ranging from 9% at station RMGR to 35% at station DJISU, most of these stations being located in urban areas (population over 10000 inhabitants). However, for most of the stations, the drops in seismic noise vary in the 10-20% interval. As the frequency increases, so does the noise level variation. In Band 2, fifty-one stations show noise reduction varying from 5% at station IACR, located in a small commune, to 35% at the stations sited in the Unirea Hotel in Focsani (Vrancea area). However, most of the stations located in larger urban areas reveal noise reductions in two intervals, 10-20% and 20-30%, respectively. In Band 3, the number of stations showing the noise reduction increases to fifty-seven, and the drops in seismic noise vary from 6% at the station TGMR in Targu Mures to 66% at the station BPLR in Bucharest. We observe the largest drops in noise levels in Band 4. For the stations GSMB and BDTR located in Bucharest the noise decreased over 80%. Large values were also observed at other stations in Bucharest or near the city as well as at the station ADJ located in the Adjud city. In total, fifty-four stations show noise reduction in Band 4. From Figure 2 we also observe that the reduction of noise is weaker for the stations in less populated and remote areas. At these stations, the drop in seismic noise varies between 10 and 20%, except for the station CJR where the noise decreased by 34% in Band 4 during the lockdown.

Analysis of the data over a longer time-period allowed us to compare the COVID-19 related noise changes with those observed during Orthodox celebrations (Easter and Christmas) as well as summer and winter holidays. During these periods human mobility and activities decrease considerably compared to normal working-days. In Figure 3, we show the evolution of the DRMS at 4 stations during the March 2019 - September 2020 period for different frequency bands. The stations are installed in different locations: DJISU (Figure 3a) is sited in the yard of the Inspectorate for Emergency Situations (IES) in Craiova city (population > 250,000) close to the main national road and railway; CTISU (Figure 3b) is installed at the basement of the IES building in Constanta city (population > 270,000) close to the main and secondary streets; PMGR (Figure 3c) is located close to Bucharest in the park of Mogosoaia town (population ~ 8,000), near important tourist attractions (the Palace of Mogosoaia and the church); and PMB1 (Figure 3d) is a station located downtown Bucharest (population > 1,800,000) in the City Hall building, near a very busy boulevard.



All stations located in large urban areas (CTISU, DJISU, PMB1) show clear drops in seismic noise between working-days and weekends and during the religious and winter holidays (Orthodox Easter 2019, Christmas 2019 and New Year 2020). The

seismic noise at stations CTISU and DJISU is affected at lower frequencies (Bands 1 and 2) mainly by noise sources generated by light and heavy traffic as these stations are close to important roads and streets. The lowest noise level is observed during the Easter and winter holidays. For station CTISU the drop in DRMS during the lockdown is comparable to the drop observed during the holidays and reaches the minimum during the 2020 Easter (April 17-20, 2020). Alternatively, the reduction of noise at station DJISU due to quarantine measures does not reach the level observed during 2019 holidays, except for the Orthodox

Easter in 2020. At the station PMB1, noise changes in Band 4 are regular, with relatively constant large drops between working-days and weekends and during the holidays. At this station, in comparison to others, another seismic noise reduction related to a religious celebration is observed. On August 15, 2019, the Assumption is celebrated and people working for the public institutions had a day off. In 2019, this holiday fell on a Thursday, and the Romanian authorities, to help the tourism, have established that Friday will also be a day off. Therefore, the noise reduction is observed over a period of several days, which

includes the weekend. During lockdown the noise level at PMB1 drops significantly, reaching the level observed during Orthodox holidays and even exceeds it, at the time of Easter in 2020. Station PMGR, on the other hand, shows a different trend. First, the seismic noise increases during the weekends, when more people go out in the park for recreational activities and for visiting the palace and church, and decreases on working-days, when there are fewer people walking in the park. Second, the noise reduction is obvious only for the Christmas 2019 and New Year 2020 holidays, when temperatures in

Romania are around 0 degrees Celsius and not many people go for a walk in the park. During the warmer holidays (Orthodox Easter and Assumption 2019), the noise level increases and reaches its maximum level. The lowest level of the noise at this station is, however, reached during the lockdown.

### 3.2 Station in cities

In this section, we present the results for several stations located in urban environments in different contexts: free field, in

schools and in buildings at different floors.

### 3.2.1 Free field-stations

In urban areas, road traffic, the underground and surface transportation system (tram, train) as well as industry represent the most important noise sources responsible for generating high-frequency anthropogenic vibrations (Long, 1971; Green et al., 2017). These sources are different from one city to another and even can vary within cities. The preventive measures taken to

limit the spread of the COVID-19 within the communities have affected all the above-mentioned noise sources in urban areas. Even though the lockdown was uniformly imposed at the national level, the reduction in seismic noise shows significant variability among the stations located in the same cities (Figure 2). The largest reductions in seismic noise were observed in Band 3 and Band 4 for the station located in urban areas, as follows: values up to 75% at ADJ (Adjud city), 66% at BPLR



(Bucharest city) and 52% at PMGR (Mogosoaia city). In Band 1 and Band 2, the largest drops of 35% and 31% were noticed
at stations DJISU (Craiova city) and TRGR (Targoviste city), respectively.

We show the results for several stations deployed in different urban conditions. We first display the noise changes at the station
BSTR, sited downtown Bucharest in one of the busiest areas of the city. The station is close to the two main boulevards and
roundabouts, with heavy traffic generated by cars and buses, and also located about 100 m away from the metro stations. The
noise at this site is very high and is generated by all of these sources. At this station, we found a reduction in seismic noise
after the lockdown in all frequency bands, with a maximum of 27% observed in Band 4. The noise level starts to decrease after
the school closure on March 11, 2020 and reaches its minimum after the stay-at-home order (Figure 4a). In the same frequency
range, the noise reduction is comparable with the one seen during the Orthodox celebrations. However, the reduction in noise
in Band 1 is similar for the Orthodox Easter in 2019 and 2020. The 24-hour clock plots in Figure 4b, show similar patterns
before and after lockdown started, denoting constant noise sources in both periods, but less intense during the lockdown. In
Band 1, the noise variation is uniform with no variation between working-days and weekends. The restriction of night activities
during the lockdown is responsible for reduction of the noise level observed during the night hours before lockdown. In Band
4, the noise variations show signatures related to signals generated by the subway transportation system (Diaz et al., 2017).
The subway train running schedule is between 5 a.m. and 11 p.m.  - the last train leaving from any terminal station - for both
working-days and weekends. The frequency of the trains increases during morning hours, decreases slowly at noon and then
increases again until 7 p.m., when the number of trains in use starts to decrease again (see the trains schedule,
http://www.metrorex.ro/program_circulatie_in_zile_lucratoare_p1379-1). During the weekend the frequency of trains remains
constant throughout the work program. We noticed a good correlation between the noise variation and the schedule of the
subway trains. For the working-days, we identify two lobes of maximum amplitude, in the morning and in the afternoon,
respectively. In between, we observe a decrease in the noise level, which is more pronounced between 11 a.m. and 1 p.m. and
is related to the higher interval between trains. In contrast, there is no variation in seismic noise during the weekend. After the
lockdown starts, the pattern remains almost the same as before, but the noise level is reduced.

The station BPLR is also located in Bucharest close to one of the largest shopping centers in the city. Other significant noise
sources affecting the site are three main boulevards with intense traffic, one close college and possibly, the subway station,
which is about 500 m away. The observed reduction in seismic noise is up to 66% in Band 3 and up to ~42% in Band 4. We
link the decrease of the noise amplitude in these frequency bands to the reduction of school activities and less to those of the
shopping center. The seismic noise level dropped (by up to 66%) following the school closure on March 11, 2020, and stayed
low, at the level observed during religious and winter holidays in 2019, until the end of lockdown (Figure 5). The shopping
centers were reopened to the public in mid June 2020 and this moment increased the noise observed in July. After the quarantine
law, the noise level decreased and increased again to reach its maximum after the lockdown, when the schools were reopened
in September 2020.

Another relevant example is shown by the CTISU station, located in Constanta city, in the basement of the IES building. In
the station area, there are also other public institutions, such as the General Inspectorate of Gendarmerie of Constanta County,



General Directorate of Social Assistance and Child Protection as well as two educational units, a secondary school and a college. The reduction in seismic noise following the lockdown is visible in all frequency bands (Figure 6). In Band 1 and

Band 2, we see a very homogeneous variation in seismic noise before and after lockdown. We observe an increase in the noise level in the morning hours and a relatively constant level between 9 a.m. and 4-5 p.m. The noise level decays in the evening hours. Before lockdown, we see an increase in the noise levels during weekends' night hours, which is no longer observable after lockdown. We assume that the pattern of the noise variation in Band 2 is related to the overall traffic in the area, while Band 1 shows a combination between the overall traffic and the specific activities at the IES. In Band 3 and Band 4, we observe

before lockdown 2 peaks of the noise levels related to the arrival and departure times of employees to and from work. The lockdown has significantly reduced the number of employees working in the office, and this reduction translates into the disappearance of the 2 peaks. In Band 3, we continue to observe an increasing trend in the noise levels during night hours and we assume that the traffic still has a contribution in this frequency domain.

### 3.2.2 Stations in schools

The pandemic has strongly impacted the education process in many countries around the world, including Romania. The first measures taken by the Romanian authorities were to close all schools (from kindergarten to high schools) in the country starting with March 11, 2020. The universities were closed on March 24, 2020, when the stay-at-home order was given. These moments were captured very well by 5 stations of RSN installed in educational units (BDTR, BVES, CBBR, GSMB, SGEB) and 2 stations sited in the proximity of schools (ADJ and BPLR). All of them show the largest drops in seismic noise of up to 80%

in Band 4, except for station BPLR (see aforementioned discussion). In Figure 7a, we show the long-term evolution of the noise DRMS for a station located in kindergarten in Bucharest (BDTR) and one sited in University Babes-Bolyai of Cluj-Napoca city (CBBR). In both cases, the noise reduction is recognized immediately after the school's closure. For station BDTR, the drop in seismic noise is steep and up to 80%. The noise level reaches the level observed during the 2019 religious (Easter and Christmas), summer and winter holidays. The noise level remains low until the mid of September 2020, when students

and teachers return to school. Station CBBR shows a slightly different behavior with a drop of 46% during the lockdown. The noise level gradually decreases after the school closure and reaches the level observed during holidays at the time of the lockdown. The minimum is reached during the 2020 Orthodox Easter, after which the noise level begins to increase and reaches a maximum for two weeks at the end of July 2020. The noise level drops again to the lockdown level and increases at the end of September 2020 when students start returning to face-to-face learning. At these stations, the noise is predominantly

generated in Band 4 by human mobility and activity inside the buildings where the stations are located. Figure 7b highlights this observation and shows the noise variation for each day of the week for the period before and after lockdown, in a 24-hour clock representation. The noise level at the BDTR station is influenced by the time marks specific to the teaching activities conducted in a preschool education unit. It increases with the arrival of children to the kindergarten (at 8 a.m.), reaches a maximum around 10–11 a.m. when the educational process occurs, decreases during the rest period (between 1 and 3 p.m.)

and increases slightly again when children are picked up by their parents between 3:30 and 4:30 p.m. This pattern fully


disappears during the lockdown. At the CBBR station, the educational activities within the university are much more uniform throughout the day, and it translates into noise variation without significant fluctuations. After the lockdown started, the noise shows a similar pattern as before, but with a clear and significant reduction in amplitude.

### 3.2.3 Stations in buildings used for structural monitoring

In this section, we present the results of the changes in the seismic noise observed at three seismically instrumented buildings. One is located downtown Bucharest, one in Magurele, a town (located ~ 15km south from Bucharest) and one in the city center of Focsani (Vrancea area). The Bucharest City Hall building (PMB) was retrofitted in 2016 and equipped with earthquake-protection system (base-isolators and viscous dampers) and the monitoring system consisting of 4 accelerometers inside the structure at the ground floor (PMB1), 2nd floor (PMB2), 3rd floor (PMB3) and attic (PMB4). The second building, located in

Magurele city, is the headquarter of the Institute of Atomic Physics (IAP). It is an office building retrofitted after the Vrancea 1977 earthquake and instrumented with 3 accelerometers installed at the basement (TURN1), 6th floor (TURN2) and 10th floor (TURN3, see Tiganescu et al., 2019; 2020). The third instrumented building is Unirea Hotel in Focsani where stations are deployed at the basement of the structure (FOCR1), the 4th floor (FOCR2) and the 8th floor (FOCR3).

Stations located in the Bucharest City Hall building show similar behavior in terms of DRMS changes, depending on the

frequency band in which they were analyzed and regardless of the floor on which the stations are installed (Figure 8). The lockdown effect is observed in all frequency domains, but is more pronounced in Band 4, with noise reductions between 41 and 49% and in Band 3, with noise reductions between 23% and 27%. In both frequency domains, the noise starts to decrease after the school closure on March 11, 2020. Variations of the noise, with maxima during working-days and minimum during weekends, are visible in the lockdown period. The lower noise levels are comparable with those observed during the religious

and winter holidays in 2019. The noise level gradually increases before the lockdown is lifted, and after the state of alert is declared it reaches the noise level observed before the lockdown. In Band 3, the seismic noise drops again starting at the end of August 2020 to the noise level observed during the lockdown. This drop is associated with the start of the campaign for the local elections in Bucharest. It is worth mentioning here the increase of the noise level between floors and the differences in the noise level between frequency bands. The noise levels increase 5 times at the station at the top of the building compared

to the station at the ground level in Band 3 and about 10 times in Band 4. However, the noise level for the stations on the same floor is 2–3 times larger in Band 3 than in Band 4. This behaviour could be attributed also to the structural peculiarities of the building and the sensor position. The E shape masonry structure is a rigid construction, with thick walls (ranging from 42 cm for the interior walls of upper stories up to 112 cm exterior walls at the basement) and reinforced concrete slabs. We should mention that all the sensors are located above the seismic protection system, thus the high-frequency content of the exterior

vibration sources is reduced, as indicated in Figure 8 for Band 3 and Band 4. However, for the upper floors, the high-frequency vibrations are transmitted and amplified by the structure itself.

In Band 1 and Band 2 the lockdown effects are not as visible as in the higher frequency bands. In Band 1, the noise level decreases gradually starting with March 11, 2020, until the end of September 2020 when it becomes comparable with the level





observed during the Orthodox Easter, Christmas and New Year 2019. In Band 2, the variation of the noise is reduced and
relatively constant over the entire analyzed period of 19 months. Starting with the end of August 2020 a significant drop is
observed. This decrease in the noise level is associated with the start of the campaign for the local elections.  In these two
frequency domains, the variations of the noise level between stations at different floors and between the Band 1 and Band 2
are not as pronounced as for the higher frequency bands.

In Figure 9, we show in a 24-hour clock representation the noise variation at station PMB1 in Band 1 to Band 4 for each day
of the week for the period before lockdown and after lockdown started. In Band 1, and to some extent in Band 2, the lockdown
has filtered out the contribution of different sources on the noise variation. Before the lockdown, the noise behaviour is similar
between working-days, with the noise level starting to increase around 5 a.m. until 8 a.m. Between 8 a.m. and 4 p.m., the noise
level is relatively constant and afterwards, it starts slowly to decrease until 10 p.m. when the reduction accelerates. During
night hours, the noise level is minimum, except the weekend when we observe an intensification of the noise level between 12
p.m. and 3 a.m. We also see an increase in the noise during Saturdays between 8 a.m. and noon compared to Sundays. During
the lockdown, the noise increases more steeply to its maximum around 8 a.m. and then decreases more rapidly after 4 p.m.
The decrease of the noise level between 12 a.m. and 2 p.m. is more pronounced than before lockdown. Furthermore, the noise
attains the same level during the Saturdays and Sundays after the lockdown and no increase in the noise level is observed
during the night. In Band 2, the pattern variation is similar before and after the lockdown. Before lockdown, we remark a
sharper increase in the noise level starting around 4:30 a.m. after which it reaches its maximum around 6 a.m. During the
lockdown, the noise starts to increase at the same hour as before lockdown, but its amplitude becomes maximum only after 8
a.m. In Band 3 and Band 4, we again observe the same pattern in noise changes before lockdown and after the lockdown
started. It is worth mentioning the sharp increase in the noise level to its maximum in the morning, between 5 a.m. and 8 a.m.,
as well as its decay starting with 3:30 p.m. during working-days. This noise variation is associated with the people coming and
going from work at City Hall.

The reduction in noise is observed for the IAP building only at the stations deployed on the 6th and 10th floor and is stronger
in Band 4. The seismic noise level dropped after the lockdown 59% at the station on the 6th floor and 62% at the station at the
top of the building (Figure 10a). The noise reduction started a few days after the school closure and extended even after the
lockdown was lifted and the state of alert imposed. The noise started to increase in June 2020, but the level before the lockdown
was not reached by the end of September 2020. Figure 10a also shows that the Orthodox Easter 2019 and the
Christmas/NewYear 2019 holidays are quieter than the lockdown period, except at the time of Orthodox Easter 2020 when the
noise levels are similar. From the 24-hour clock representation of the noise variation (Figure 10c, d), we noticed similar
behavior for the two stations before and during the lockdown associated with the work schedule of the people working inside
the building. It is interesting to note here that the activity in the building lasts longer on Wednesdays and it reduces sooner on
Fridays, being a six-hours working day, before the lockdown. After the lockdown, this pattern is no longer recognized.

The pandemic has also impacted human activities within hotels. The seismic noise decreased at the stations deployed in the
building of Unirea Hotel located in Focsani city by ~35% in Band 2 and ~23%-29% in Band 3 following the lockdown. In





Band 1 and Band 4 noise reductions were also observed, but they were much weaker. In Figure 11, we choose to present the results only for the station deployed on the last floor of the hotel (FOCR3), as for the other 2 stations the results are similar.

The seismic noise started to decrease with the closing of the schools on March 11, 2020, and remained at the lowest level between the stay-at-home and the state of alert orders. After that, the noise level quickly returned to the pre-pandemic level. The 24-hour clock representation (Figure 11) shows changes in the noise variation pattern before and after lockdown. In the first case, the noise level remains high during the working-days until 8 p.m., after which it starts to decrease. A slight increase in the noise level can be observed starting with 9 o'clock in the evening for Fridays and Saturdays, which can be associated

with a prolongation of human activity during the weekend nights. After the lockdown, the noise level drops after 5 p.m. for working-days and after 2 p.m. for Saturdays and Sundays, and the weekend nights remain as quiet as during the week nights.

### 3.3 Stations in less populated areas

We observed a weaker noise reduction at six sites with populations less than 2000 inhabitants (Figure 2). The long-term evolution of the DRMS is shown in Figure 12 for two selected remote stations. The first one, station Cheia-Muntele Rosu

(MLR), is part of the auxiliary seismic network of the International Monitoring System (IMS) installed in a vault in a remote setting and is one of the quietest seismic stations of RSN (Grecu et al., 2012). The noise reduction due to the lockdown is up to 24% in Band 1 and 18% in Band 2, respectively. The noise level reaches a minimum during the Orthodox Easter 2020 and increases afterwards toward a slightly higher level than the one observed during summer and beginning of autumn 2020. We also observe a seasonal variation in seismic noise, showing a gentle increase during warmer months followed by a decrease

over the colder months. In the area, there are 2 cottages where tourists used to come for recreation. The larger cottage is about 400 m away from the vault, while the smaller one about 200 m away from it. We would have expected to detect increases in seismic noise during holidays or weekends when the area is much more crowded than on working-days, but it was not observed in the variation of the noise. Instead, the noise level decreases during the weekend and increases during the working-days. Therefore, we assume that the noise variation is mainly related to the human activity around the seismological observatory at

Muntele Rosu, which is about 70 m away from the vault. The noise peaks observed in the long-term evolution of the DRMS are also related to increased winds in the area (Mihai et al., 2019, see Figure S2 from SI).

The second station is located on a plateau on the Feleacului Hill, far from the city of Cluj-Napoca, in the courtyard of the Astronomical Observatory of the Romanian Academy, Cluj-Napoca Branch, about 1.5 km away from the national road DN1 and almost 1 km from the nearest residential houses. The observatory is not open to the public and mainly the staff working

there is using the building. The area is a well-known place for walks and hikes for many residents of Cluj-Napoca city, several mountain bike trails existing also in the area. As a consequence, we assume that the noise sources responsible for the long-term noise level at station CJR combine the noise generated by people going to work at the observatory and by people doing outdoor activities in the area. We observed that the peaks of the long-term noise changes are associated with either the weekends or the working-days, which suggests that the weather conditions play also a role in the variations in the seismic

noise. IT is likely that when there is good weather more people come in the area for outdoor activities and the noise level



during weekends exceeds the noise level generated by the staff working at the observatory. An interesting aspect worth mentioning here is the low level of the seismic noise observed in the first weekend of August 2019. We assume that this particular noise drop is associated with the largest international music festival in Romania (UNTOLD), hosted in Cluj-Napoca. During the festival many residents of the city preferred to attend the festival rather than going for outdoor activities in the area

of CJR station. The noise drop associated with the lockdown is observed a few days after the stay-at-home order and reaches a minimum during the Orthodox Easter 2020. The noise level is comparable with the noise level observed during the UNTOLD festival or Christmas 2019 and remains relatively low until the lockdown is lifted, after which it starts to increase constantly until the end of September 2020.

## 4 Discussions

In densely populated and industrialized areas, high-frequency seismic noise wavefield is characterised by the superposition of signals of many different anthropogenic origins (Denolle and Nissen-Meyer, 2020). These sources generate elastic waves in a wide range of frequency bands and also vary in time and space, making it often difficult to discriminate between different noise sources at stations in cities. However, recent studies that have focused on the analysis of urban seismic noise variations provided insights into the sources responsible for generating seismic noise in different frequency bands. Road traffic has an

important contribution to the seismic noise wavefield in various frequency bands, such as 2-9 Hz, 10-20 Hz (Dias et al., 2020), 2.5-10 Hz (Green et al., 2017), 8-35 Hz (Boese et al., 2015), 8-12 Hz (Diaz et al., 2017). Overground and subway trains generate seismic noise at very low frequencies (~0.01 - 0.05 Hz) (Sheen et al., 2009; Diaz et al., 2017; Green et al., 2017) as well as at high frequency (20-40 Hz) (Diaz et al., 2017). Furthermore, industrial activities can also contribute to the seismic noise spectra in the 1-25 Hz and 25-40 Hz bands (Groos and Ritter, 2009).

The measures taken to prevent and combat the spread of COVID-19 have impacted many human and industrial activities across Romania and their effects are clearly emphasized by the recordings of many stations of RSN. It is very difficult to quantify the level of disruption of anthropogenic activities and in addition, society's response differs from one area to another making it thus difficult to achieve a generally unerring interpretation of the results. However, our analysis of lockdown related seismic noise reduction performed in four frequency bands highlights a number of common features. Clear homogeneous patterns of

noise reductions have been observed in Band 1 (Figure 2a). Most of the stations showing noise reduction in this frequency band are sited within larger urban areas (with population more than 10000 inhabitants) and are close to streets. The stations in Bucharest provide a suggestive example of recordings where the seismic noise is sensitive to the traffic. The variation of noise reduction is the smallest at these stations, closely reflecting their proximity to heavily-trafficked main streets. An additional argument for the connection between the seismic noise (in Band 1) and the traffic is given by the decrease of the noise level

during the summer vacation when this is considerably reduced in Bucharest (Figure 4a). The noise reduction during the lockdown period is smaller than that observed during the Orthodox Easter and Christmas 2019, suggesting that the traffic during the lockdown did not decrease as much as during the two major holidays. The reduction of traffic during the pandemic





allowed to highlight the contribution of other sources to the noise spectrum in Band 1. For example, during the lockdown period we could observe at the PMB stations (within Bucharest City Hall) a sharp variation of the noise level associated with

the working schedule of the people coming and going from work, i.e., a sharp increase in the noise level starting with 8 a.m and a decrease starting with 4 p.m. (Figure 9b).

Groos and Ritter (2009) suggested that industry noise is important in the 25-40 Hz frequency band for the stations in Bucharest. However, we found that the noise reduction in Band 4 due to the lockdown measures is mainly related to the restrictions on human mobility and activities close to the area of the station site. We observed the largest drops in seismic noise levels at

stations in schools. These drops are of the same level as those observed during the school's vacations, (including summer and winter holidays, the Orthodox Easter and Christmas). In addition, the analysis of the noise variation according to the time and days of the week shows different patterns before and during the lockdown (Figure 7c). If in the first situation the pattern is clearly given by the activity carried out inside and near the school, in the second one it disappears completely, suggesting the complete interruption of the activity in and around the school. Similar behaviour is found also for stations installed a few tens-

hundred meters away from schools. Another category of stations that show significant noise reductions in Band 4 are those located in buildings. For these stations the noise reduction during the lockdown is associated with the mobility restriction of the people working inside the building.

For the stations in Bucharest close to metro stations (e.g., BSTR) we associated the noise reduction with changes in the schedule of underground trains during lockdown.

The contribution of noise sources such as traffic, human movement to the noise spectra in Band 2 and Band 3 depends on the type of the dominant sources and the distances to these sources. Figure 6a and Figure 12b show clearly two different cases. For the station CTISU (Figure 6a), located within an area with many streets with intense traffic, the noise reduction in Band 2 during the lockdown period is associated with a reduction in the traffic around the station location. On the other hand, for the station CJR (Figure 12b), located in a remote area known for walks and outdoor activities and far from the city traffic, the

noise reduction during the lockdown is mainly due to the human movement in the vicinity of the station site. At this station a minimum level of seismic noise was also observed during an important music festival taking place in Cluj-Napoca. We assume that the noise was reduced due to the fact that many inhabitants of Cluj-Napoca chose to go to the festival and less to go for a walk and outdoor activities in the area of the CJR seismic station. Another good example for the influence of the movement of people in the vicinity of the station on seismic noise is shown by the PMGR station (15-25 Hz, see Figure S3 from SI). The

station PMGR, located in a park, shows a clear dependency of the noise level in Band 3 on the human movement within the park. The park is a promenade place preferred by the inhabitants of Bucharest and which at the end of the week is very crowded. As a consequence, the noise level increases during the weekends in comparison with the working-days. During the lockdown, when outdoor social activities were restricted, the noise dropped significantly for all the days of the week. However, people were allowed to go to the park for sports activities (e.g., running) during the lockdown, and the noise peaks observed on

Saturday and Sunday mornings during the lockdown are related to such outdoor sports activities.





To provide some insights into the causes of the noise changes during the lockdown, correlation between noise time series and community mobility data provided by Google and Apple was performed globally (Lecocq et al., 2020a) and at local scale (Cannata et al., 2021; De Plaen et al., 2021; Diaz et al., 2021). The results showed rather good similarities between the noise and mobility data, indicating that the changes in seismic noise can be used to track the human activity in urban areas. We also

observed a good match between the variation in seismic noise and data mobility's trends (Figure S4), confirming thus the previous results (Cannata et al., 2021; De Plaen et al., 2021; Diaz et al., 2021).

The high level of seismic noise in cities affects the earthquake detection capability at seismic stations installed in these areas and as a consequence, fewer small earthquakes are recorded by these stations in comparison with those deployed at similar epicentral distances but in quieter environments. Therefore, any reduction in anthropogenic seismic noise improves the

detection capability of urban stations (Lecocq et al., 2020a), and such an increased detectability would translate into more data for research on seismic risk reduction in earthquake-prone cities. The reduction of seismic noise during the Romanian lockdown led to an improvement in earthquake detection for the urban strong motion instruments of RSN. The low noise level during the lockdown allowed the detection with enough accuracy of a moderate (ML=3.8) Vrancea intermediate-depth (~116 km) earthquake at the accelerometers sited in urban areas. For a local event of this size, the anthropogenic noise energy usually

masks the earthquake signals, and thus it would be difficult to identify the event on the urban accelerograms without any filtering applied (Figure 13).

## 4 Conclusions

The permanent seismic stations operated by the RSN provided a very valuable data set for investigating the variation of seismic noise during the period associated with the reduction of human activities caused by COVID-19. Our analysis for stations

located in different regions of the country as well as in various contexts shows that noise reduction is more important at stations located in urban areas where the contribution of anthropogenic noise sources is dominant in the noise spectrum. However, we found drops in lockdown-related seismic noise even at stations located in remote areas where anthropogenic activity is much reduced.

Even though the lockdown has been imposed uniformly nationwide, our investigation in four different frequency bands reveals

substantial variability in seismic noise reduction both among the stations, even for those located in the same city, and frequency bands. The results show the greatest reductions in seismic noise in places where people's movement has been severely affected by the restrictive measures taken because of COVID- 19. We found such large drops - over 40% and up to 80% - in and near educational units as well as in buildings in the 15-40 Hz frequency range. The level of noise in these situations is similar to that observed during other periods when human mobility and social activities reach a minimum, such as religious celebrations

or school vacations.

We found that the contribution of other noise sources, such as traffic, is important in the 2-14 Hz frequency range. However, the noise reduction during the lockdown period due to traffic is more uniform among urban stations (Figure 2a) and is less

pronounced than in the higher frequencies, up to 35%. In these cases, the seismic noise level was reduced less than during the religious celebrations, suggesting that the traffic during the lockdown period wasn't so much diminished in comparison with
the days off associated with Easter and Christmas 2019.

Our results finally reveal that noise reduction caused by the measures taken to mitigate the COVID-19 pandemic can influence the seismic monitoring of local events, by improving the detection capability of stations in noisy urban environments. Therefore, we consider it is essential to continue the efforts to reduce the seismic noise in the seismological data acquired by urban stations, as these efforts would lead to the improvement of seismological databases used for seismic risk reduction in
earthquake-prone cities.

**Code availability**

This work has benefited from open-source initiatives such as Obspy (Krischer et al., 2015) and QGIS -A Free and Open Source Geographic Information System (https://qgis.org). Data analysis has been done using the publicly available SeismoRMS code kindly distributed by Thomas Lecocq (Lecocq et al., 2020b).

**Data availability**

Part of the seismic data used in this study are available from the FDSN service, provided by the Romanian Seismic Network of the National Institute for Earth Physics: https://doi.org/10.7914/SN/RO. For the stations that are not available through FDSN, the authors have used data provided by the NIEP's Data Center and are available upon request. The mobility data for Bucharest were provided by Apple (https://covid19.apple.com/mobility, last access: February 12 2021) and for all the large
cities in Romania by Google (https://www.google.com/covid19/mobility, last access: February 12 2021). The authors are very grateful to Dr. I. Moldovan from NIEP for providing the meteorological data recorded at Muntele Rosu.

**Author contribution**

BG designed the study and all the authors contributed to the manuscript. The data processing was performed by all authors, as follows: BG and DT processed the local accelerometric data from urban areas, NP processed the data available through FDSN,
AT processed the data in schools / buildings, RD and FB processed the local free-field seismic data. BG, FB, AT and NP interpreted the results and drew the conclusions.

**Competing interests**

The authors declare that they have no conflict of interest.



**Special issue statement**

This article is part of the special issue "Social seismology – the effect of COVID-19 lockdown measures on seismology". It is not associated with a conference.

**Acknowledgements**

Part of the work has been supported by the following national funded projects: Open system for integrated monitoring of civil structures (PREVENT), supported by a grant of the Romanian Ministry of Research and Innovation, UEFISCDI, project

number PN-III-P2-2.1-PED-2019-0832, within PNCDI III; National Core Funding Program (NUCLEU) program Multidisciplinary Program research on the seismic phenomenon in order to increase resilience (MULTIRISC), supported by Ministry of Education (MEC), project number PN1908020. Another part of the work has been supported by the SETTING project (Integrated thematic services in the field of Earth System Observation - a national platform for innovation), cofunded from the Regional Development European Fund (FEDR) through the Operational Competitivity Programme 2014-2020,

Contract No. 336/390012. We thank the people involved in the development of the ObsPy package and the SeismoRMS code for sharing their work.

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





**Figure 1:** Map showing the distribution of seismic stations in Romania. The inset map shows the sensors distribution in Bucharest city. The station codes are given only for the stations mentioned within the study. VR denotes Vrancea seismic zone.





**Figure 2:** Percent change in DRMS during the period March 25-April 23, 2020 (right after the stay-at-home order entered into force) with respect to the interval February 10 - March 10, 2020 in the frequency bands a) 2- 8 Hz (Band 1); b) 4-14 Hz (Band 2); c) 15-25 Hz (Band 3) and d) 25-40 Hz (Band 4).

565





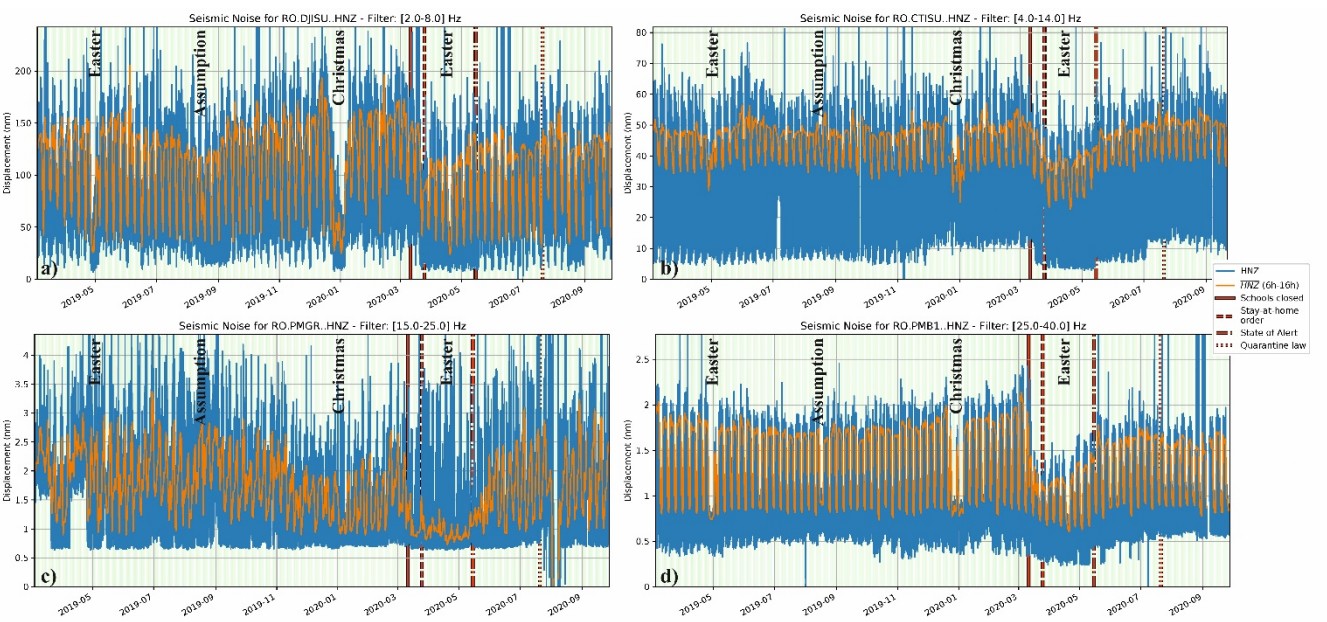

**Figure 3:** Long-term evolution of DRMS in four different frequency bands observed at the stations: a) DJISU (2-8 Hz); b) CTISU (4-14 Hz); c) PMGR (15-25 Hz) and d) PMB1 (25-40 Hz). The locations of the stations are displayed in Figure 1.





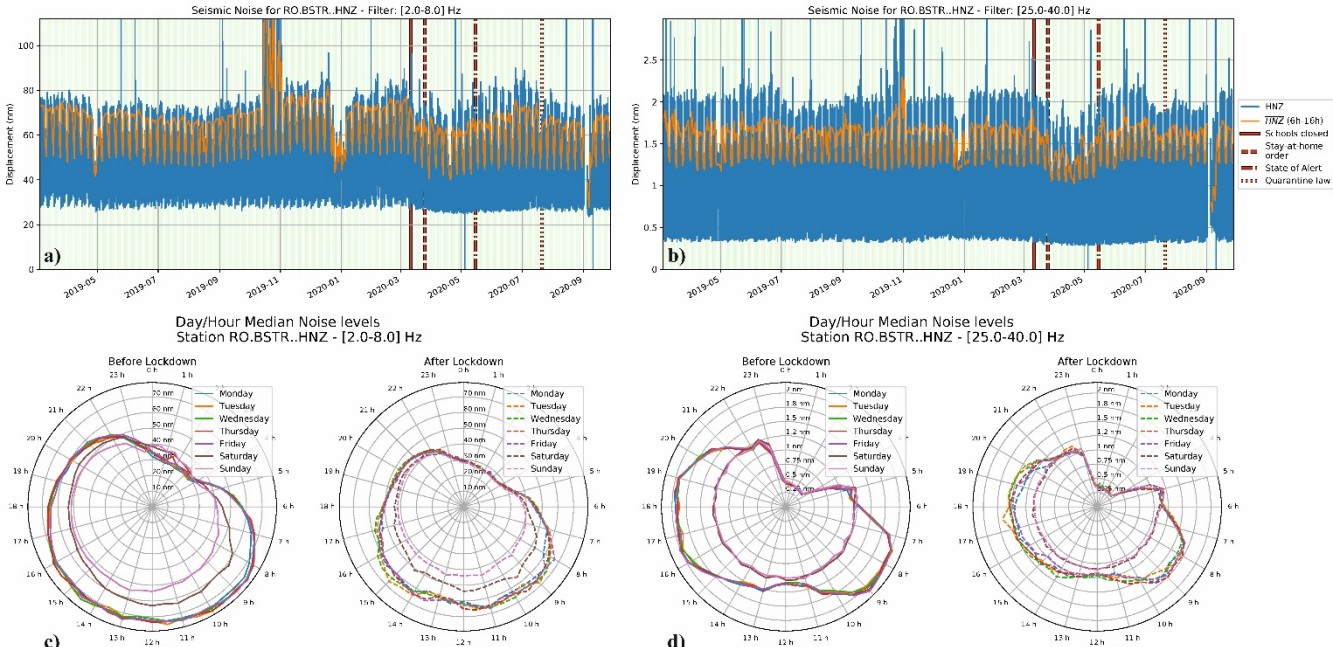

**Figure 4:** Lockdown effects on seismic noise in downtown Bucharest (BSTR): top - Evolution of DRMS for the March 2019 - September 2020 period based on displacement data in the bands 2-8 Hz (a) and 25-40 Hz (b); bottom - 24-hour clock plots showing average displacement variation for each day of the week and for the period before lockdown and during lockdown for the bands 2-8 Hz (c) and 25-40 Hz (d). The location of the station is displayed in Figure 1.

595





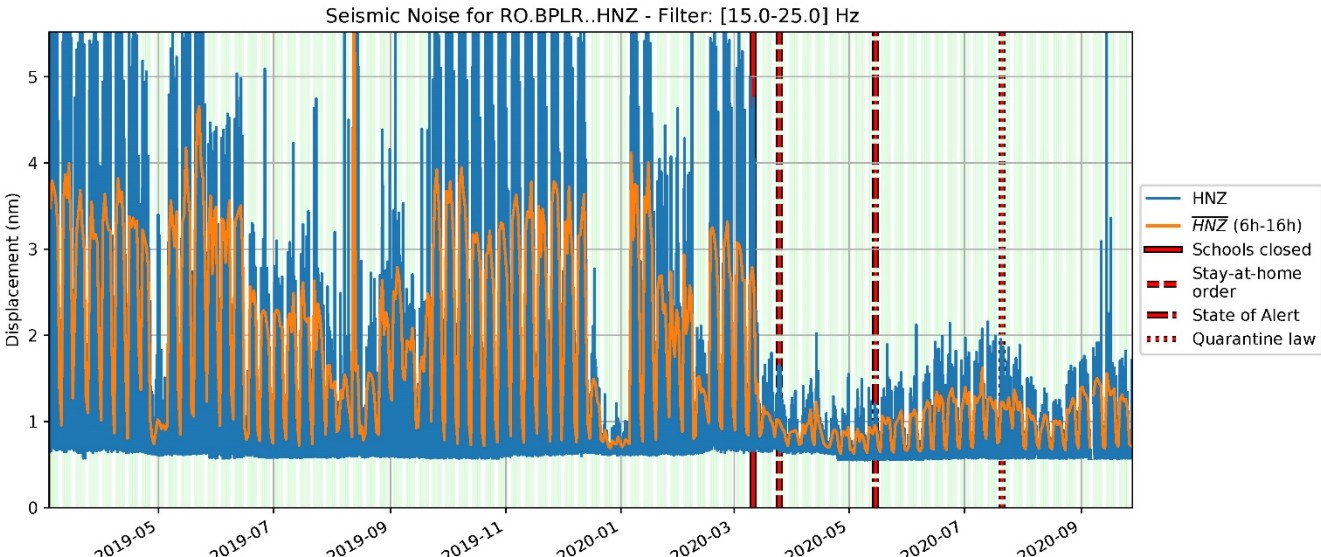

**Figure 5:** Lockdown effects on the seismic noise at station BPLR in Bucharest. The large drop in seismic noise of up to 66% is observed right after the school closure on March 11, 2020 (see Figure 1 for the station location).





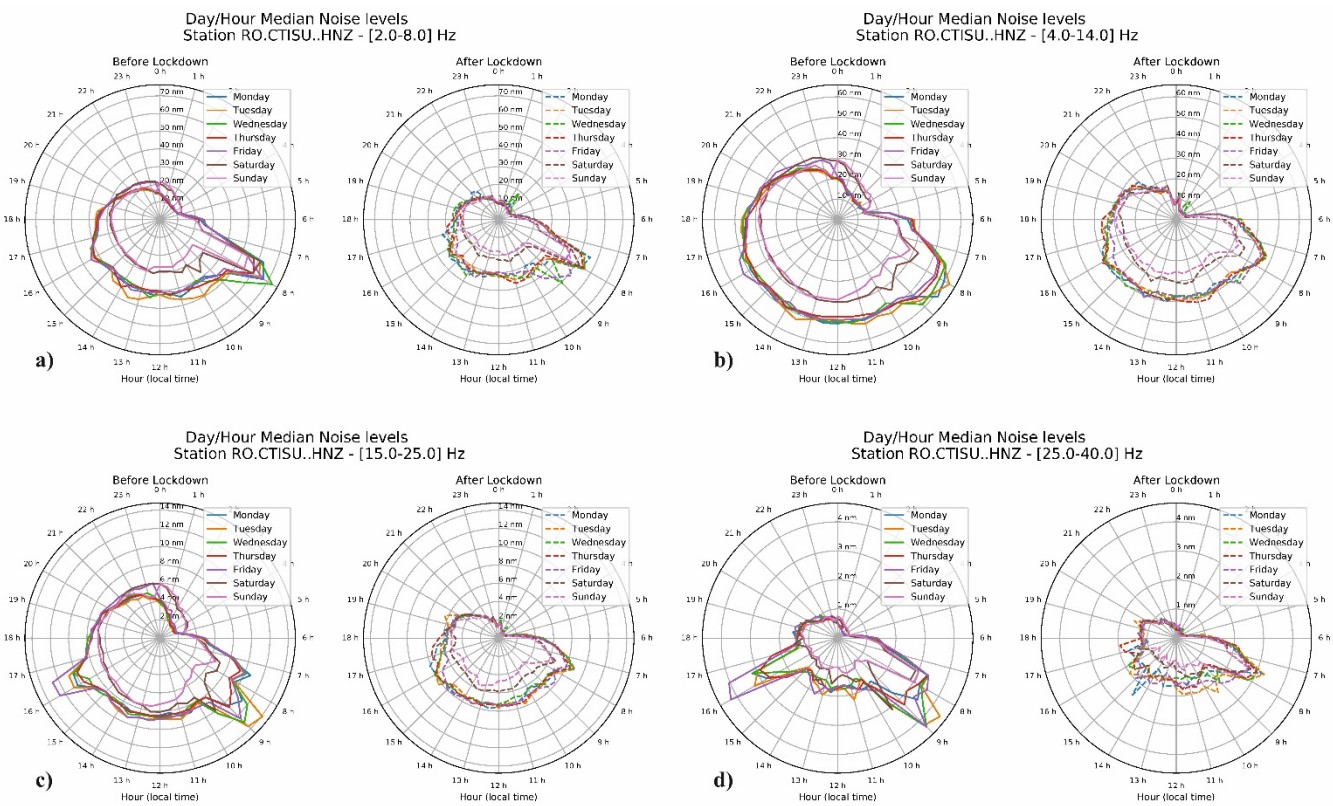

**Figure 6:** Lockdown effects shown on 24-hour clock plots at station CTISU in Constanta city for the frequency bands 2-8 Hz (a), 4-14 Hz (b), 15-25 Hz (c) and 25-40 Hz (d, see Figure 1 for the station location).





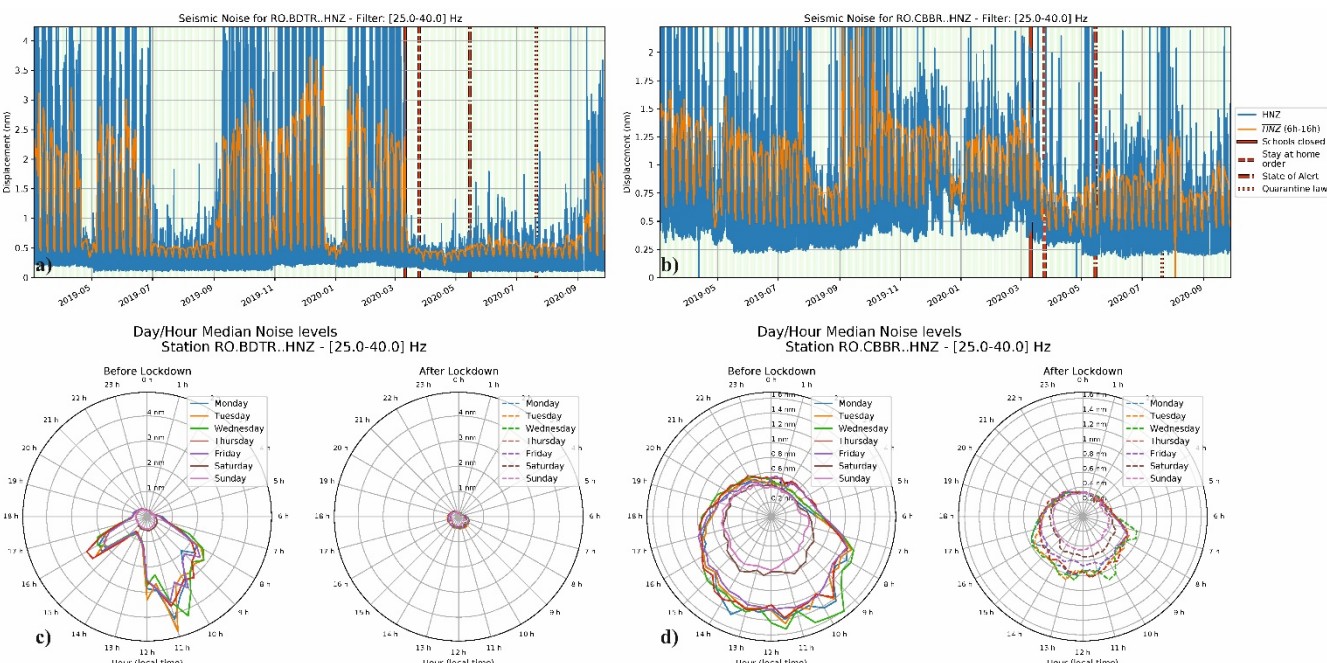

**Figure 7:** Temporal changes in seismic noise at a kindergarten in Bucharest (BDTR) (a) and university in Cluj-Napoca (CBBR) (b). 24-hour clock plots showing average displacement variation for each day of the week and for the period before and during lockdown at c) BDTR and d) CBBR, stations. The locations of the stations are displayed in Figure 1.









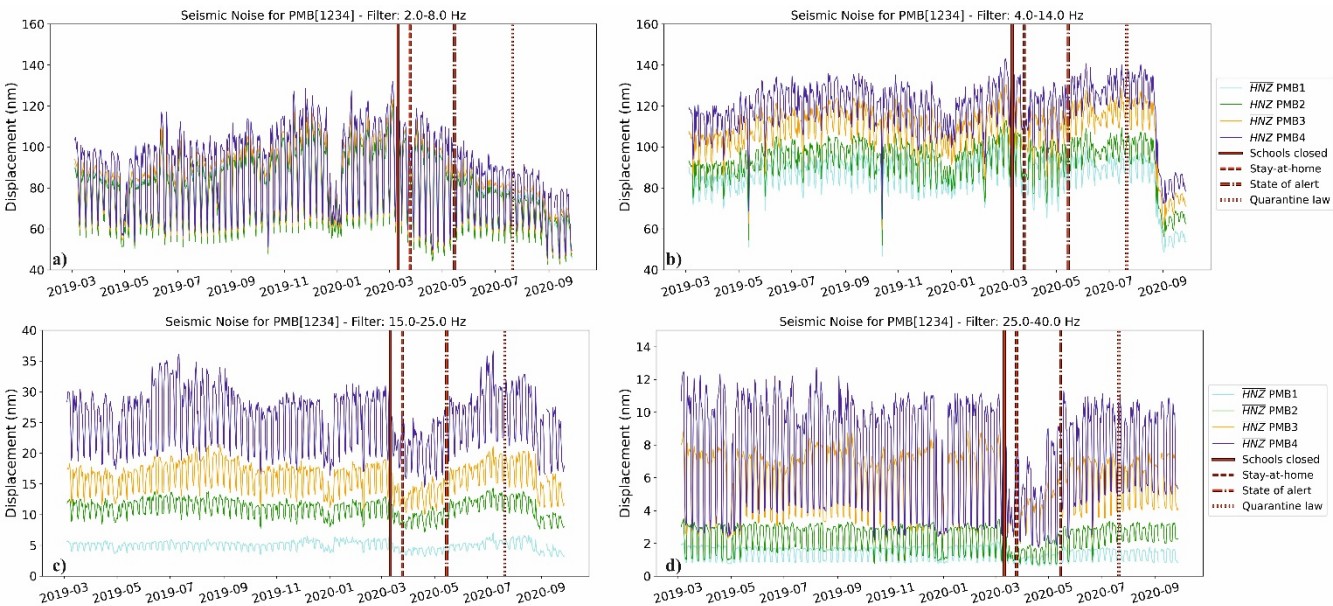

**Figure 8:** Long-term DRMS variations at the stations located in the Bucharest City Hall building observed in the frequency bands 2-8 Hz (a), 4-14 Hz (b), 15-25 Hz (c) and 25-40 Hz (d). Note the higher noise levels in the lower frequency bands as well as the significant increase of the noise levels at the stations deployed from the ground floor to the top of the building (see Figure 1 for the station location).









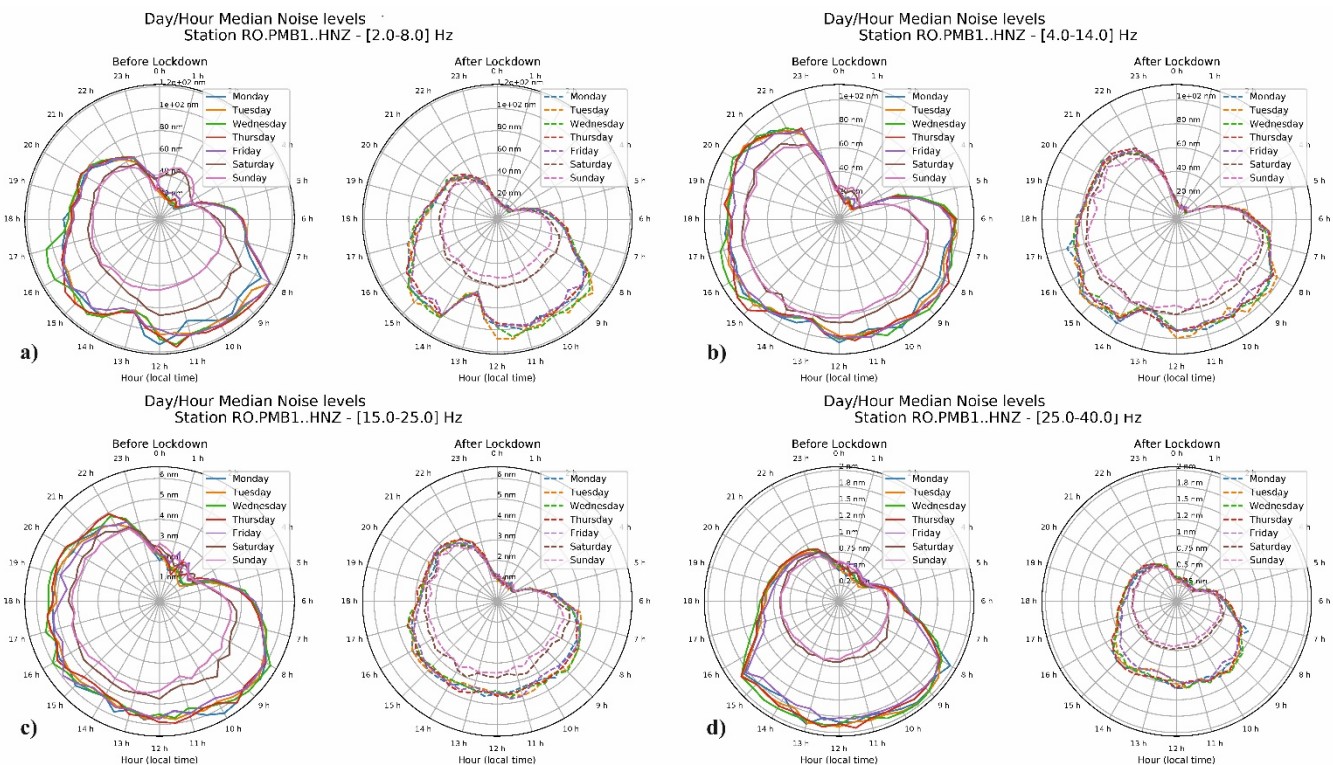

**Figure 9:** Lockdown effects shown on 24-hour clock plots at the station PMB1 located in the Bucharest City Hall building (ground floor) for the frequency bands 2-8 Hz (a), 4-14 Hz (b), 15-25 Hz (c) and 25-40 Hz (d, see Figure 1 for the station location).








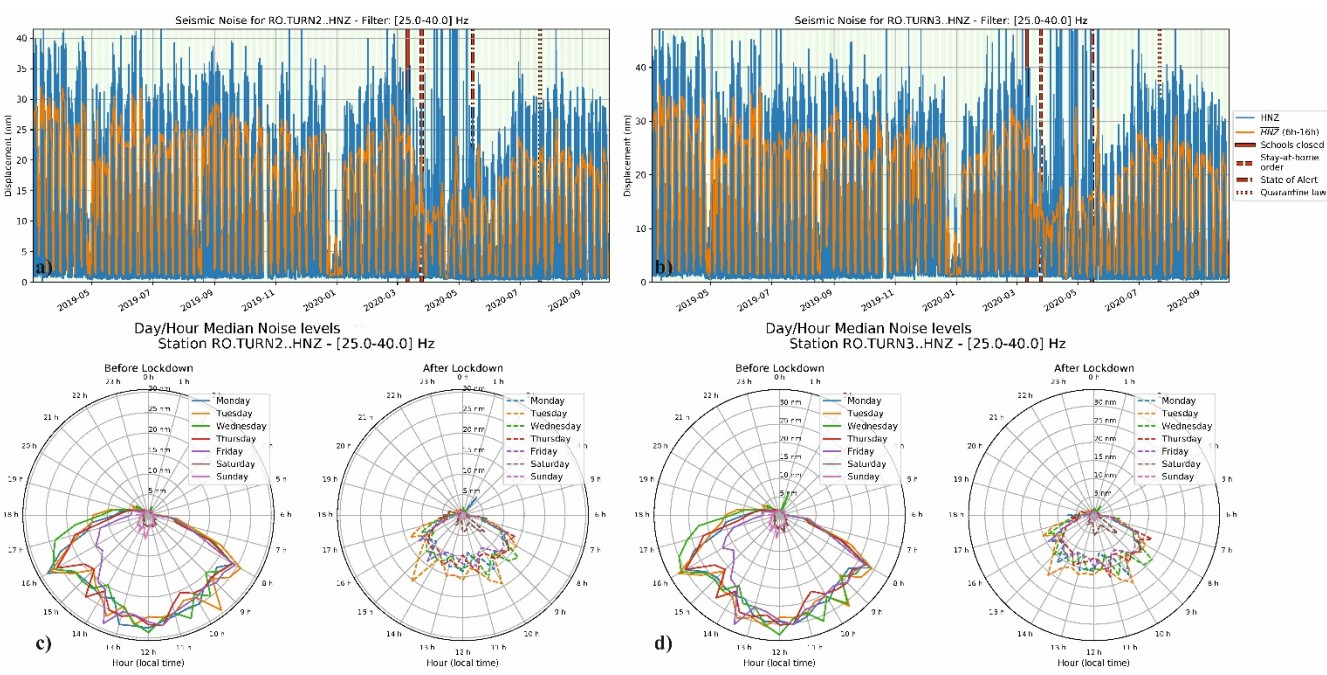

**Figure 10:** Temporal changes in seismic noise observed at the station located on the 6th (a) and 10th (b) floors of the IAP building. Lockdown effects are shown on 24-hour clock plots for both stations in (c) and (d, see Figure 1 for the station location).





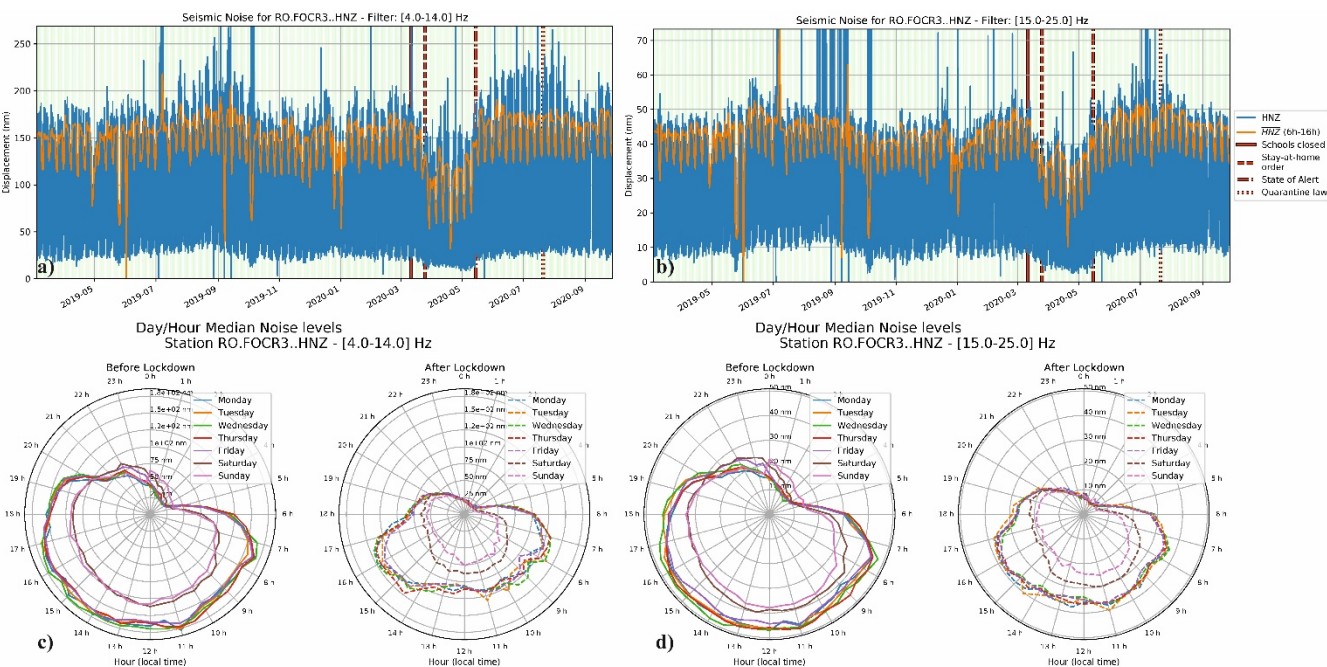


**Figure 11:** Temporal changes in seismic noise observed at the station located on the 8th floor of the Unirea Hotel in Focsani in the frequency bands 4-14 Hz (a) and 15-25 Hz (b). Lockdown effects are shown on 24-hour clock plots in the frequency bands 4-14 Hz (c) and 15-25 Hz (d, see Figure 1 for the station location).






**Figure 12:** Long-term changes of seismic noise at MLR (a) and CJR (b) stations located in remote places less populated. Note the comparable minimum noise level during the UNTOLD Festival and Orthodox Easter or Christmas holidays at CJR station. The locations of the stations are displayed in Figure 1.



**Figure 13:** Illustration of the increased earthquake detection capability at two stations of RSN: a) - map showing the epicenter locations of two intermediate-depth Vrancea earthquakes (2017-08-03, ML=3.8, H=117 km - red star and 2020-04-18, ML=3.8, H=118 km - blue star) and the location of the two accelerometers, in Bucharest (BUC) and Galati (GISR) cities; b) and c) - waveforms recorded by the two stations (red trace for the event before the lockdown and blue trace for the event within the lockdown); d) and e) temporal changes in seismic noise at the stations GISR and BUC.