# Peer review of "The effect of 2020 COVID-19 lockdown measures on seismic noise"

_Solid Earth, 2021_

## Author Response (AR1)

Dear Editor,

We would like to thank you for taking the time to consider our manuscript entitled "The effect of 2020 COVID-19 lockdown measures on seismic noise recorded in Romania" [manuscript no. se-2021-38] for publication in Solid Earth. Please find, in the submission section of the authors, our final detailed response to the comments received from the two reviewers. We have worked step by step through all the issues that have been raised, as outlined in the response below listing reviewers comments in black and our corresponding replies highlighted in red. Our responses are in red with new manuscript text in bold italics.

**Reviewer 1**

The manuscript "The effect of 2020 COVID-19 lockdown measures on seismic noise recorded in Romania" by Grecu and others analyze the seismic noise variation in seismic recordings from the stations of the Romanian Seismic Network, before and during the COVID-19 pandemic and lockdown periods.

Generally, the manuscript is well-written and easy to follow. The results are interesting, and, in my opinion, the manuscript can be published after some minor revisions.

We thank the reviewer for his/her careful and detailed review and positive comments.

I have the following main questions:

1. Have you also considered frequency bands lower than 2 Hz? Is there a reason why you chose to start from 2-8 Hz frequency band?

The entire data set was also analyzed in the 0.5-1 Hz frequency band, both for broadband stations and accelerometers. In the case of broadband stations, we observed clear seasonal variations of seismic noise. In contrast, the results obtained for the accelerometers do not reveal any relevant seismic noise variation at these low frequencies. An example is given in Figure 1 below, for a station that has both broadband velocity and acceleration sensors. As it can be seen from these graphs, the long-term evolution of the seismic noise is very different for the two recordings.

The long-term seismic noise-variation obtained for acceleration and velocity sensors becomes comparable starting with frequencies higher than 1 Hz, as shown in Figure S1 in the supplementary material. These aspects, and the fact that most stations used in the analysis are equipped with the acceleration sensors, led us to limiting the analysed lower frequencies to 2 Hz.

[Figure]

Figure 1. long-term variation of the noise at VRI station for the broadband velocity (top) and accelerometer (bottom) sensors. Note the seasonal variation observed for the broadband sensor and no baseline noise changes in case of the accelerometer.

2. Have you detected signals related to weather conditions during the lockdown period? Maybe at those stations where the noise signal is low like the kindergarten station, just after the school's closure.

We did not look at such signals for stations in cities, this is mainly because we did not have access to the data from weather stations installed in the cities or close to our seismic stations to correlate with. However, when we looked to see if any earthquakes were recorded during the lockdown period, we checked the waveforms and didn't notice any unusual signals. For the MLR station, where we have a collocated weather station, we were able to observe a correlation between the increase in seismic noise and wind speed increase at certain periods of time (see Figure S2 in supplementary material).

3. Line 294: here you are considering the station deployed at the last floor of the hotel. So, aren't you observing a behaviour like in Figure 8, with different noise levels at different floors?

We observed partly the same behaviour as in Figure 8, i.e. an increase in the level of noise between the basement station (FOCR1) and the station deployed at the 4th floor (FOCR2). However, the difference in the noise levels between the station at the top of the building (FOCR3) and FOCR2 are less significant. In addition, the overall characteristics of the long-term noise variations are similar regardless of the floor at which the seismic station is located.

4. Figure 11b: Around August 2019 and October 2019, there is an increase in the noise level. Any thoughts/interpretations about these peaks?

The increase in the noise level you are referring to is observed during 4-8 July 2019 (186-189 Julian day) and 11-14 September 2019 (254-257 Julian day). We computed the spectrograms (Figure 2 below) for these time periods and noticed an increase in noise energy for frequencies higher than 15 Hz. This increase is seen only during the daytime and it is stronger for the station located at the 8th floor (FOCR3), while disappearing for the stations located in the basement (FOCR1). We can only speculate about what caused this increase in noise level - most likely some works carried out on the upper floors of the hotel. Unfortunately, we have no conclusive data or information about such very local noise sources in the building.

[Figure]

Figure 2. Spectrograms computed for stations FOCR3 (8th floor), FOCR2 (4th floor) and FOCR1 (basement) for the time interval of 30 June 2019 and 14 July 2019 (Julian days 181-195). Note the increase of the noise power at station FOCR3 for frequencies >  15 Hz and Julian days 185-189.

Minor-technical points:

Overall, I suggest increasing the size of the font for the clock plots. Even enlarging, I still found it difficult to read letters and numbers.

We increased the size of the font for the clock plots.

Can you please add the holidays (e.g., Orthodox Easter) in those Figures where they are missing (like Figure 7)?

We added the missing labels (Easter, Christmas) to Figure 7 and Figure 1.1

Abstract, Line 9: March 2020

Done

Line 325: typo, IT

Done

**Reviewer 2**

General Comments:

The manuscript of Bogdan Grecu and co-authors examines the effect of COVID-19 lockdown measures on seismic noise recorded by the Romanian Seismic Network.

I have to note that quite similar observations of seismic noise reductions were recently documented by a considerable number of studies on this topic.

However, in my opinion, a "country scale" analysis, like the current one, and the corresponding observations regarding the changes of seismic noise levels in relation to the Romanian measures against COVID-19, deserve to be published. I congratulate the authors for the significant volume of data analyzed.

I recommend the manuscript for publication in Solid Earth's special issue on "Social Seismology and the effect of COVID19 lockdown measures on seismology", after making the following suggested adjustments.

We thank the reviewer for his/her careful review and positive comments.

Main Comment:

My most significant comment concerns the "earthquake detection capability" part of the manuscript. In the abstract, the authors state that in the framework of their current analysis, they show that noise reduction during the lockdown has also improved the earthquake detection capability of the accelerometers located in noisy urban environments.

However, a potential reader of the manuscript must reach the last four lines of the "Discussion Section", before he can get some information about this topic.

Besides that, the discussion/information which is provided about the "pre-lockdown" and "post-lockdown" earthquakes, is quite limited to adequately support a reliable conclusion regarding the "improvement in earthquake detection capability".

What are the exact origin times of the two earthquakes (day of the week and local time of occurrence)? Only the dates are provided.

We added the origin time in the Figure 13's caption. In case of the first earthquake (2017-08-03) the origin time is 13:13:16 (local time) while for the second event (2020-04-18)  the origin time is 19:17:03 (local time)

Assuming that the search I have made is proper, the "pre-lockdown" earthquake occurred on Thursday, 2017-08-03 13:13 (local time), while the "post-lockdown" earthquake on Saturday, 2020-04-18 19:17 (local time). In case that the above-mentioned origin times are correct, I believe that such a comparison is not quite fair and it possibly leads to misleading conclusions.

If the authors agree, I would recommend perhaps to totally exclude the part of "earthquake detection capability" from the manuscript, considering also that the structure of the paper will be slightly affected in such case.

Although we agree to some extent with the reviewer's comments, pointing that it is difficult to draw a general conclusion about the detection capability of seismic stations in cities based on the observations from only two earthquakes, we would like, however, to keep the "earthquake detection capability" analysis part within the Discussion section. We do believe  that the seismic event of 18 April 2020 was clearly (with low SNR) recorded due to the seismic noise reduction during the lockdown. To support our statement, we provide the 24-hour clock plots for the two stations mentioned in the Discussion section and shown in Fig. 13 . Plots for both stations show the reduction of the noise during the lockdown for weekdays and weekends. We consider that these noise level drops lead to an overall higher detection capability and the time of the event occurrence (13:13 and 19:17 local time) most likely have only a second order effect as being less significant.  This characteristic was also outlined by other studies (Lecocq et al., 2019 - https://doi.org/10.1126/science.abd243; Pérez-Campos et al. - https://doi.org/10.5194/se-12-1411-2021, 2021.)

[Figure]

[Figure]

Figure 3. The effect of lockdown on 24-hour clock plot representations at stations BUC (top) and GISR (bottom).

To take into account reviewer's comment we removed from the abstract sentence referring to "earthquake detection capability" and modified the corresponding sentence of the Conclusions to: "*Our results finally reveal that noise reduction caused by the measures taken to mitigate the COVID-19 pandemic may indicate a potential improvement in the earthquake  detection capability of the accelerometers located in noisy urban environments*".

Specific Comments:

Abstract

L09: "March 2019"→ "March 2020"

Done

L12-13: "containing 148 stations" → I would recommend using the phrase "consisting of 148 stations".

Done

L14: To be more precise, the reduced human activity is mostly due to the lockdown measures and not due to COVID-19 in general. I would suggest rephrasing that part accordingly. For example, "...in Romania due to COVID-19" → "...due to the Romanian measures against COVID-19".

Done

L15: "corresponds to" → "correspond to"

Done

L18: "In the lower frequency range (2-8 Hz and 4-14 Hz) the variability of the noise reduction among the stations is lower than in the high frequency range, and the noise level is reduced by up to 35%.". I find this sentence a bit confusing, especially in the context of an abstract. Could you please clarify and maybe rephrase it? In addition, no information about the percentage reduction observed at higher frequencies (15-40 Hz) is provided.

We consider that the sentence is not unclear if taken in full context, as noted below. Within the abstract we provided the level drops for the both low and high frequency ranges ( 2-14Hz and 15-40Hz). However, to acknowledge the comment, we made an attempt to re-formulate the last part of the corresponding sentence. We hope that the modified version is more clear and easier to follow.

"We focused our investigation on four frequency bands - 2-8 Hz, 4-14 Hz, 15-25 Hz and 25-40 Hz  and found that the largest reductions in seismic noise associated with the lockdown corresponds to the high frequency range of 15 - 40 Hz. We found that all the stations with large reductions in seismic noise (> ~40%) are located inside and near schools or in buildings, indicating that at these frequencies the drop is related to the drastic reduction of human activity in these edificies. In the lower frequency range (2-8 Hz and 4-14 Hz) the variability of the noise reduction among the stations is lower than in the high frequency range, corresponding to about  35% on average."

1 Introduction

L46: "The study analyzed noise data..." → I would recommend rephrasing this part. E.g., "In the latter study, seismic noise data were analyzed..."

We changed the text to: "*In this study, seismic noise data from more than 300 stations distributed worldwide were analyzed and the results pointed out ...*"

L47: "are responsible" → I would recommend writing "were responsible"

Done

L49: "..., the first official case in the country being reported on..."→ "..., with the first official case being reported on..."

Done

L57: "all movement was" → I would recommend writing "all movements were"

Done

**2 Data and method**

L75: "within the medium to large urban areas" → "within medium to large urban areas"

Done

L88: " DRMS'" → Please remove the " ' ".

Done

L84-86: "We choose the above frequency intervals taking into account different contributions that the anthropogenic noise sources have in a wide frequency range, starting from 0.02 Hz up to 40 Hz (Sheen et al., 2009; Boese et al., 2015; Diaz et al., 2017)."

This sentence could raise some potential questions. For example, why low frequency seismic noise (< 1 Hz) was not included in the analysis?

The actual reason due to which the frequency analysis was limited up to 40 Hz, is that the frequency content of the anthropogenic noise is strictly limited up to such frequencies? Does the choice of the specific frequency range depend on the sampling rate of the data? It would be good to include some extra comments about your choice of concentrating on the specific frequency bands (2-8 Hz, 4-14 Hz, 15-25 Hz and 25-40 Hz).

As replied to Reviewer 1 (comment 1), we did investigate the variation of seismic noise in the 0.5-1 Hz frequency band for the seismic stations equipped with broadband velocity sensors as well. In this low frequency band we observed only seasonal variations of the noise and no changes related to Covid-19 restrictions. The upper frequency limit of 40 Hz was chosen based on the examples from the literature.. In addition, we took into account that the sampling rate of the data is 100 Hz, which limits the frequency analysis band to 50 Hz. We also included some arguments related to the considered frequency ranges in the Discussions section.

To take into account the reviewer's comment we added to the text the reviewer is referring to "... - more details in the Discussions section). In addition, in order to avoid the seasonal variations of seismic noise at low frequencies (0.1-1 Hz), we chose to perform our analyses starting from 2 Hz. The upper limit of 40 Hz was adopted following the numerous example of previous studies (e.g., Groos and Ritter, 2009; Diaz et al., 2017)."

**3 Results**

**3.1 General overview**

L99: "computed the median of the noise DRMS": The temporal variations of DRMS presented in the totality of the manuscript's figures, is superimposed with the temporal variation of the median DRMS during working hours (6h-16h). Did the computation of the median DRMS

values computed for the two 30-days long time intervals follow the same approach? This should be clear in the manuscript.

Yes, the median DRMS values are computed for the two 30-days long time intervals following the same approach. We modified the text accordingly: "*... computed the median of the noise DRMS during working hours (6h-16h) for two 30 day intervals,...*".

L98-100: "we computed the median…", ..."We compute the percentage...":I would suggest not shifting tenses between sentences.

Done. We chose the past tense "We computed the percentage"

L101-103: "For each site, we represented a circle colored according to the maximum percentage of the noise reduction in each band and sized as a function of the number of inhabitants in the area":I would suggest not including so much details about the color coding or the symbol size of the plots in the main text. I would just refer to the overall content of Figure 2 (e.g., percentage change of the median DRMS for each frequency band).

We moved the sentence "*For each site, we represented a circle colored according to the maximum percentage of the noise reduction in each band and sized as a function of the number of inhabitants in the area*" to Figure 2's caption.

L106: "10000 inhabitants" :I would recommend keeping the same number formatting with or without a thousand separator throughout the manuscript.

Done. We chose the number formatting with a thousand separator "10,000"

L113: "Large values"→ I would recommend writing "Large seismic noise drops"

Done.

L144: "... the noise reduction is obvious…"→ I would recommend writing "... the noise reduction is evident..."

Done.

L146: "The lowest level of the noise…"→ I would recommend writing "The lowest noise level…"

Done.

3.2 Station in cities

L148: "Station in cities"→ "Stations in cities"

Done.

3.2.1 Free field-stations

L151: "Free field-stations"→ "Free-field stations"

Done.

L156: "was uniformly imposed at the national level" → "was uniformly imposed at a national level"

Done.

L158:"...for the station..."→ "...for the stations..."

Done.

L162: "The station is close to the two main boulevards…" → "The station is close to two main boulevards…"

Done.

L163-164:"The noise at this site is very high and is generated by all of these sources.": I would remove this sentence.

Done.

L168: "The 24-hour clock plots in Figure 4b…": The specific plots are labeled as 4c in Figure 4. I would also suggest including a general reference to Figure 4 describing its overall content in this paragraph.

We added in the text "*Figure 4 shows the lockdown effects on seismic noise at station BSTR.*" We also changed "the 24-hour clock plots in Figure 4b" to "...Figure 4c, d" and referenced Figure 4b once again in the text at line 174 "...of 27% observed in Band 4 (Figure 4b).".

L170-171: "The restriction of night activities during the lockdown is responsible for reduction of the noise level observed during the night hours before lockdown":This sentence is not perfectly clear to me.

We changed the sentence to: "*The lockdown resulted in the restriction of the night-time activities, which led to a reduction in seismic noise compared to that observed before.*"

L176: "During the weekend..." → "During the weekends…"

Done.

L180: "... the higher interval between trains." → Consider replacing the word "higher". E.g., "the longer inter-train intervals."

Done. We accepted the suggestion.

L183: "..., one close college…" → "a nearby college"

Done.

L187-178: "The shopping centers were reopened to the public in mid-June 2020 and this moment increased the noise observed in July."Please consider rephrasing this sentence.

We rephrased the sentence to: "*The increase of seismic noise observed in July 2020 is linked to the reopening of the shopping centers starting from mid-June 2020.*"

L188-190: "After the quarantine law, the noise level decreased and increased again to reach its maximum after the lockdown, when the schools were reopened in September 2020":The same for this sentence also. Please consider rephrasing it. Consider maybe splitting this part into two separated sentences.

We modified the phrase to: "*After the quarantine law, a slight decrease in seismic noise is observed. The noise level grew again reaching its maximum after the lockdown, when the schools were reopened in September 2020.*"

L191: Is station CTISU considered as a free-field station? It is installed in the IES's building. If yes, this part should be moved to another section.

In section 3.2.3, we took into account only the stations that are used for the structural health monitoring of the buildings in which they are deployed. Although station CTISU is installed in the basement of a one-storey IES's building, we considered it a free-field station as it is used only for seismic monitoring.

3.2.2 Stations in schools

L211: "... located in kindergarten in Bucharest…" →"... located in a kindergarten in Bucharest…"

Done.

L213-214: "The noise level reaches the level observed during the 2019 religious (Easter and Christmas), summer and winter holidays.":Easter, Christmas and other holidays are discussed in the text but they are not labeled in Figure 7. Please consider labeling the previously mentioned holidays, as you did in Figure 3, for example.

Done. We added the missing labels (Easter, Christmas) in Figure 7.

L220: "Figure 7b highlights...": The specific plots are labeled as 7c in Figure 7.

Done. Changed to 7c

3.2.3 Stations in buildings used for structural monitoring

L235: "headquarter" →"headquarters"

Done.

L281-290: No station ID(s) is/are mentioned in this paragraph. Please include somewhere in this paragraph the station IDs you are referring to (TURN2, TURN3).

At the beginning of the section 3.2.3, we mentioned in the text "... 3 accelerometers installed at the basement (TURN1), 6th floor (TURN2) and 10th floor (TURN3, see Tiganescu et al., 2019; 2020)". However, to make reading easier we added the station IDs at the beginning of the paragraph "*... IAP building only at the stations deployed on the 6th (TURN2) and 10th floor (TURN 3) ...*"

L285-286: Easter, Christmas and other holidays are discussed in the text but they are not labeled in Figure 11. Please consider labeling the previously mentioned holidays, as you did in Figure 3, for example.

Done. We added the missing labels (Easter, Christmas) in Figure 11.

5 Conclusions

L407: This section should be numbered as 5.

Done.

L410: "noise reduction is more important" →"noise reduction is more significant"

Done.

L418: "The level of noise" →"The seismic noise level"

Done.

---

## Editor Decision (ED1)

Dear Dr. Grecu and coauthors,

Thank you for submitting your revised manuscript and for addressing in detail all comments of the reviewers, particularly also clarifying the choice of frequency band used.

I have to say that I agree with the main comment of reviewer 2 about the earthquake detectability, that the comparison performed is not entirely fair. Not necessarily the time of day, but the fact that the pre-lockdown event is on a Thursday and the post-lockdown event is on a Saturday, makes a difference. Looking at the 24-hour clock plots provided, this means that the drop in seismic noise you are considering is larger than that due to lockdown (e.g. a change of 25 nm at station GISR, (from 60 nm on Thursdays 13h pre-lockdown to 35 nm Saturdays 19h post-lockdown), rather than a drop of about 5-10 nm due to lockdown for a particular time/day).

I have noted that some text was already altered following the reviewers suggestion, but I ask the authors to adapt the text in the Discussion and Conclusions further. While I agree there is an improved SNR, this is not conclusively due to the lockdown and thus requires further investigation. Perhaps rephrase the text to just mention that there is possibility for improved capability, as reported in other studies.

In addition to the description in the Introduction (third paragraph) of the lockdown measures in Romania, it would be helpful to have this in a Table perhaps, as aid to the reader.

Finally, I have given the manuscript a read-through myself and I have some minor technical comments (mostly rewording of text) as listed below. I would be grateful if you could make these changes in your revised version of the manuscript as well (which I estimate would not take much time). I hope these changes are clear as I could not refer to line numbers.

Given the comments listed above, I recommend that your manuscript is accepted after these minor corrections are done.

Thank you again for your manuscript, and wishing you a good remainder of the summer.

Best wishes,

Paula Koelemeijer

**Minor technical comments:**

- Consider rephrasing the use of "the COVID-19" everywhere to just "COVID-19" or "the COVID-19 pandemic" if appropriate.

**Abstract:**

- Consider changing the phrase "mobility and activity in Romania due to the Romanian measures against COVID-19" to " mobility and activity due to the Romanian measures against COVID-19"

**Introduction:**

- First paragraph, page 1: "seismic noise has natural origin" → "seismic noise has a natural origin,"

- Second paragraph, page 2: "an unprecedented disruption in anthropic activities in many cities around the globe"

- Consider removing "in many cities" as I would think activities were affected everywhere, not just in cities.

- Second paragraph, page 2: "coronavirus disease (COVID-19) and having a direct effect on seismic noise recorded by seismic stations." → "coronavirus disease (COVID-19), having a direct effect on seismic noise recorded by seismic stations."

- Third paragraph, page 2: "and starting with May 15, 2020 gradual relaxation measures (opening of some shops, museums, etc.) were resumed" → "and from May 15, 2020 gradually some activities (opening of some shops, museums, etc.) were resumed"

- Third paragraph, page 2: "still needed for limiting the spread of the COVID-19" → "are still needed for limiting the spread of COVID-19"

**Data and methods**

- Third paragraph, page 3: "We choose the above frequency intervals taking into account different contributions that the anthropogenic noise sources have in a wide frequency range"
Consider rephrasing to "We choose the above frequency intervals to take into account different contributions from anthropogenic noise sources in a wide frequency range"

- Fourth paragraph, page 3: "only strong motion instruments and in addition"

Consider starting a new sentence before "In addition", e.g. "only strong motion instruments. In addition"

**Results**

- Section 3.1, second paragraph: Consider using numerals for numbers larger than 10, to help readability.

- Section 3.1, fourth paragraph: "reaches the minimum during the 2020 Easter (April 17-20, 2020). Alternatively, the reduction of noise" → "reaches the minimum during the 2020 Easter weekend (April 17-20, 2020). In contrast, the reduction of noise"

- Section 3.1, fourth paragraph: "during 2019 holidays, except for the Orthodox Easter in 2020."
Do the authors mean the 2019-2020 holidays?

- Section 3.2.1, third paragraph: "After the quarantine law, a slight decrease in seismic noise is observed" → After the quarantine law came into place, a slight decrease in seismic noise is observed"

- Section 3.2.3, second paragraph: "The noise level gradually increases before the lockdown is lifted, and after the state of alert is declared it reaches the noise level observed before the lockdown". Do the authors mean to say, "after the state of alert is lifted, it reaches"?

- Section 3.2.3, second paragraph (and repeated in the third paragraph): "This drop is associated with the start of the campaign for the local elections in Bucharest." Why does the start of this campaign lead to decreased noise levels? Could the authors please clarify this?

- Section 3.2.3, fourth paragraph: The authors describe in the pattern of seismic noise variations in a lot of detail, and mention what causes the variations in Band 2. I wonder whether the authors could add some statements on what causes the variations in Band 1 as well, or is this the same?

- Section 3.2.3, last paragraph: "The seismic noise started to decrease with the closing of the schools on March 11, 2020, and remained at the lowest level between the stay-at-home and the state of alert orders". Are there schools located in the vicinity of the hotel or is there another reason the seismic noise starts to decrease when the schools closed?

- Section 3.3, second paragraph: "mainly the staff working there is using the building." → "mainly the staff uses the building"

- Section 3.3, second paragraph: "several mountain bike trails existing also in the area" → "with several mountain bike trails existing also in the area"

- Section 3.3, second paragraph: "at station CJR combine the noise" → "at station CJR are a combination of the noise"

**Discussion**

- Second paragraph, page 11: "their effects are clearly emphasized by the recordings of many stations of RSN" → "their effects are clearly observed in the recordings of many stations of RSN"

- Fourth paragraph, page 12: "those located in buildings." → "those located in office buildings."

- Fifth paragraph, page 12: "preferred by the inhabitants of Bucharest and which at the end of the week is very crowded" → "preferred by the inhabitants of Bucharest, which gets very crowded at the end of the week"

**Data availibility**

- "the authors have used data provided by the NIEP's Data Center and are available upon request" → "the authors have used data provided by the NIEP's Data Center, which are available upon request"

**Figures**

- Temporal evolution plots of seismic noise (e.g. Fig. 3, 7, 8, 10, 11 etc): the titles in these plots are quite small. Please consider enlarging them.

- Figure 2: Would the authors please consider not using green and red symbols to denote different percentage changes, to aid the readability for readers that are colour blind?

- Figure 4: It would be helpful to add the time periods used in the clock plots in the caption

---

## Author Response (AR2)

Dear Editor,

We would like to thank you for taking the time to read our manuscript entitled "The effect of 2020 COVID-19 lockdown measures on seismic noise recorded in Romania" [manuscript no. se-2021-38] and provided useful suggestions to raise the quality of the paper. We have worked step by step through all the issues that have been raised, as outlined in the response below listing your comments in black and our corresponding replies highlighted in red.

Thank you for submitting your revised manuscript and for addressing in detail all comments of the reviewers, particularly also clarifying the choice of frequency band used.

We also thank you and both reviewers for their careful and detailed review.

I have to say that I agree with the main comment of reviewer 2 about the earthquake detectability, that the comparison performed is not entirely fair. Not necessarily the time of day, but the fact that the prelockdown event is on a Thursday and the post-lockdown event is on a Saturday, makes a difference. Looking at the 24-hour clock plots provided, this means that the drop in seismic noise you are considering is larger than that due to lockdown (e.g. a change of 25 nm at station GISR, (from 60 nm on Thursdays 13h pre-lockdown to 35 nm Saturdays 19h post-lockdown), rather than a drop of about 5-10 nm due to lockdown for a particular time/day).

We admit that the subject of earthquake detectability is rather complex involving many features (eg. seismic source, radiation pattern, seasonal and hourly noise level variation, etc). However, by analysing the seismic data recorded pre and post-lockdown periods we emphasized significant noise drops especially in the high-frequency range for many of the selected stations. Although, in our example, as you correctly noticed the difference in noise variation is not so meaningful, this was the most representative earthquake pair, having similar sizes and source characteristics.

I have noted that some text was already altered following the reviewers suggestion, but I ask the authors to adapt the text in the Discussion and Conclusions further. While I agree there is an improved SNR, this is not conclusively due to the lockdown and thus requires further investigation. Perhaps rephrase the text to just mention that there is possibility for improved capability, as reported in other studies.

To address the comment, we changed the paragraph in the Discussion sections as follows:

"The reduction of seismic noise during the Romanian lockdown could also favour an increase in the earthquake detection capability. Figure 13 shows examples of unfiltered accelerograms (from two sensors sited in urban areas) of two moderate (ML=3.8) intermediate-depth (~116km) earthquakes from Vrancea. One of the earthquakes occurred before the lockdown, on 2017-08-03 (Thursday), 13:13:16 local time, and the other during the lockdown on 2020-04-18 (Saturday), 19:17:03 local time. It is worth noting that seismic signals are clearly recorded for the earthquake generated during the lockdown despite that for a local event of this size, the anthropogenic noise usually masks earthquake signals. Although the noise variation between working days and weekends is considerable, noise drop due to the lockdown measures contributes as well, favouring the increase of the signal-to-noise ratio. In this context, the reduction of seismic noise during the lockdown may lead to an improvement in earthquake detection for RSN accelerometers located in urban areas as was also reported by other studies (Lecocq et al., 2020a). This topic, however, requires further, more in depth investigation that is out of scope for the present study."

In addition to the description in the Introduction (third paragraph) of the lockdown measures in Romania, it would be helpful to have this in a Table perhaps, as aid to the reader.

We introduced the requested table (see below) in SI material.

| DATE | ACTION | OBSERVATIONS |
| --- | --- | --- |
| February 26, 2020 | The first case of COVID-19 was reported in Romania | |
| March 11, 2020 | All schools in Romania were closed. | The World Health Organization (WHO) declared the novel coronavirus (COVID-19) outbreak a global pandemic. |
| March 16, 2020 | The state of emergency was declared. | |
| March 17, 2020 | The first military order was issued. | Banned all outdoor activities, the closure of cafes and the restriction of the number of people in outdoor activities to a maximum of 100 persons. |
| March 21, 2020 | The second military order was issued. | Led to the closure of all shopping centers, banning of groups of more than 3 people in the streets during daytime and imposed the curfew from 10 p.m. to 6 a.m. |

| March 24, 2020 | The national lockdown law came into force. | All movements were restricted, except for work purposes, health needs and essential activities. |
|---|---|---|
| May 14, 2020 | The lockdown ended. | |
| May 15, 2020 | The activities were gradually resumed. | |
| July 18, 2020 | The quarantine law came into force. | |

Finally, I have given the manuscript a read-through myself and I have some minor technical comments (mostly rewording of text) as listed below. I would be grateful if you could make these changes in your revised version of the manuscript as well (which I estimate would not take much time). I hope these changes are clear as I could not refer to line numbers.

Given the comments listed above, I recommend that your manuscript is accepted after these minor corrections are done.

Thank you for the positive and detailed comments. We included all of the requested changes as you suggested.

Minor technical comments:
- Consider rephrasing the use of "the COVID-19" everywhere to just "COVID-19" or "the COVID-19 pandemic" if appropriate.

We have rephrased the use of "the COVID-19" throughout the text.

Abstract:
- Consider changing the phrase "mobility and activity in Romania due to the Romanian measures against COVID-19" to " mobility and activity due to the Romanian measures against COVID-19".

Done.

Introduction:
- First paragraph, page 1: "seismic noise has natural origin" → "seismic noise has a natural origin,"

Done.

- Second paragraph, page 2: "an unprecedented disruption in anthropic activities in many cities around the globe". Consider removing "in many cities" as I would think activities were affected everywhere, not just in cities.

Done.

- Second paragraph, page 2: "coronavirus disease (COVID-19) and having a direct effect on seismic noise recorded by seismic stations." → "coronavirus disease (COVID-19), having a direct effect on seismic noise recorded by seismic stations."

Done.

- Third paragraph, page 2: "and starting with May 15, 2020 gradual relaxation measures (opening of some shops, museums, etc.) were resumed" → "and from May 15, 2020 gradually some activities (opening of some shops, museums, etc.) were resumed"

Done.

- Third paragraph, page 2: "still needed for limiting the spread of the COVID-19" → "are still needed for limiting the spread of COVID-19"

Done.

Data and methods
- Third paragraph, page 3: "We choose the above frequency intervals taking into account different contributions that the anthropogenic noise sources have in a wide frequency range"
Consider rephrasing to "We choose the above frequency intervals to take into account different contributions from anthropogenic noise sources in a wide frequency range"

Done.

- Fourth paragraph, page 3: "only strong motion instruments and in addition"
Consider starting a new sentence before "In addition", e.g. "only strong motion instruments. In addition"

Done.

Results
- Section 3.1, second paragraph: Consider using numerals for numbers larger than 10, to help readability.

Done. We changed to numerals for all numbers larger than 10.

- Section 3.1, fourth paragraph: "reaches the minimum during the 2020 Easter (April 17-20, 2020). Alternatively, the reduction of noise" → "reaches the minimum during the 2020 Easter weekend (April 17-20, 2020). In contrast, the reduction of noise"

Done.

- Section 3.1, fourth paragraph: "during 2019 holidays, except for the Orthodox Easter in 2020."

Do the authors mean the 2019-2020 holidays?

No, we are referring to the Orthodox Easter 2019, Christmas 2019 holidays. We rephrase the sentence as follows: "The reduction of noise at station DJISU due to quarantine measures wasn't significant since it did not reach a level similar with the one observed during the Orthodox Easter and Christmas holidays in 2019. A similar level of noise drop was noticed only during the Orthodox Easter in 2020."

- Section 3.2.1, third paragraph: "After the quarantine law, a slight decrease in seismic noise is observed" → After the quarantine law came into place, a slight decrease in seismic noise is observed"

Done.

- Section 3.2.3, second paragraph: "The noise level gradually increases before the lockdown is lifted, and after the state of alert is declared it reaches the noise level observed before the lockdown". Do the authors mean to say, "after the state of alert is lifted, it reaches"?

No, we wanted to emphasize that the noise level started to increase even before the end of the lockdown period and after the Romanian authorities declared the state of alert (immediately after the lockdown) the noise level increased reaching the same value as before the lockdown.
We rephrase the sentence as follows "The noise level started to increase again before the end of lockdown. After the Romanian authorities lifted the lockdown restrictions, and declared the state of alert, noise level reached the level of the pre-lockdown period."

- Section 3.2.3, second paragraph (and repeated in the third paragraph): "This drop is associated with the start of the campaign for the local elections in Bucharest." Why does the start of this campaign lead to decreased noise levels? Could the authors please clarify this?

We added the following sentence to the text: "We assume that the start of the political campaign for the local election led to numerous meetings with the community. Such meetings are typically held outside of the City Hall and involve many employers This diminishes the number of people and the working hours within the City Hall building."

- Section 3.2.3, fourth paragraph: The authors describe in the pattern of seismic noise variations in a lot of detail, and mention what causes the variations in Band 2. I wonder whether the authors could add some statements on what causes the variations in Band 1 as well, or is this the same?

We added the following sentences to the text: "Band 1 seems to be the most suitable to observe people's activities within the City Hall before lockdown because it better reflects their daily schedule. We could notice, before lockdown, a sharp increase of the noise during Saturdays between 8 a.m. and noon compared to Sundays. During the lockdown the noise significantly decreased on Saturdays in the same time interval. Moreover, noise variation before and after lockdown between working hours (7 a.m.-4 p.m.) and evening (4-10 p.m.) are as well better highlighted in Band 1 as compared with Bands 2 and 3."

- Section 3.2.3, last paragraph: "The seismic noise started to decrease with the closing of the schools on March 11, 2020, and remained at the lowest level between the stay-at-home and the

state of alert orders". Are there schools located in the vicinity of the hotel or is there another reason the seismic noise starts to decrease when the schools closed?

The nearest schools are located at a distance larger than 500m from the hotel. Instead, near the Unirea Hotel, are located medical, educational and cultural centers as well as the City Prefecture, the City Hall and the County Council, which have certainly reduced the activities starting with the closure of schools in Romania.

- Section 3.3, second paragraph: "mainly the staff working there is using the building." → "mainly the staff uses the building"

Done.

- Section 3.3, second paragraph: "several mountain bike trails existing also in the area" → "with several mountain bike trails existing also in the area"

Done.

- Section 3.3, second paragraph: "at station CJR combine the noise" → "at station CJR are a combination of the noise"

Done.

Discussion
- Second paragraph, page 11: "their effects are clearly emphasized by the recordings of many stations of RSN" → "their effects are clearly observed in the recordings of many stations of RSN"

Done.

- Fourth paragraph, page 12: "those located in buildings." → "those located in office buildings."

Done.

- Fifth paragraph, page 12: "preferred by the inhabitants of Bucharest and which at the end of the week is very crowded" → "preferred by the inhabitants of Bucharest, which gets very crowded at the end of the week"

Done.

Data availibility
- "the authors have used data provided by the NIEP's Data Center and are available upon request" → "the authors have used data provided by the NIEP's Data Center, which are available upon request"

Done.

Figures

- Temporal evolution plots of seismic noise (e.g. Fig. 3, 7, 8, 10, 11 etc): the titles in these plots are quite small. Please consider enlarging them.

Done. We enlarged the titles in Fig. 3, 7, 8, 10, 11

- Figure 2: Would the authors please consider not using green and red symbols to denote different percentage changes, to aid the readability for readers that are colour blind?

Done. We changed the colours (red and green) to blue and magenta

- Figure 4: It would be helpful to add the time periods used in the clock plots in the caption

Done.

---

## Author Response (AR3)

Dear Editor,

We would like to thank you for taking the time to read our manuscript entitled "The effect of 2020 COVID-19 lockdown measures on seismic noise recorded in Romania" [manuscript no. se-2021-38] and provided usefuls suggestions to raise the quality of the paper. We have worked step by step through all the issues that have been raised, as outlined in the response below listing your comments in black and our corresponding replies highlighted in red.

Thank you for submitting your new revised manuscript and for addressing my comments.

We also thank you for your careful and detailed review.

I have had a final read through of the manuscript and I have very few minor text suggestions, listed below. I am confident you will be able to make these very quickly, so I will recommend your manuscript for publication subject to technical corrections only.
Thank you for submitting your manuscript to our special issue.

Thank you for the positive and detailed comments. We included all of the requested changes as you suggested.

Minor technical comments:
Line 59: "(from the Supporting information (SI) section)" instead of "(from Supporting information (SI) section)"

Done (Line 58)

Line 92: "(see Figure S1)" instead of "(see Figure S1 from SI)".

Done (Line 91)

Line 138: change to "similar to the one observed"

Done (Line 135)

Line 251: change to "noise levels reached"

Done (Line 248)

Line 255: I assume you mean City Hall employees instead of City Hall employers?

Yes, we meant employees. We changed employers to employees. (Line 252)

Line 286: change to "are also better highlighted in Band 1 compared with Bands 2 and 3"

Done (Line 283)

Line 426: change to "Although the noise difference"

Done (Line 417)

Line 427: change to "is considerable, the noise drop due to the lockdown measures contributes as well, further increasing the signal-to-noise ratio"

Done (Lines 418-419)

Line 430: change to "out of scope of the present"

Done (Line 422)